# Rethinking Calibration for Early-Exit Neural Networks

**Piotr Kubaty** [1 2] **Filip Szatkowski** [3 4] **Grzegorz Choczyński** [1] **Eric Nalisnick** [5] **Bartosz Wójcik** [1]

## Abstract

Early-exit neural networks (EENNs) accelerate inference by allowing intermediate classifiers to stop computation once predictions are confident enough. Most methods rely on confidence thresholds for exiting, and consequently, improving classifier calibration is widely assumed to improve performance. In this work, we challenge this assumption and show that calibration alone is not sufficient for EENNs to exploit adaptive computation. To address this insufficiency, we introduce Early-Exit Failure Prediction (EEFP), which accounts for both prediction correctness and the cost of further computation. We also propose a lightweight, EEFP-motivated procedure to improve the intermediate classifiers, which can directly replace calibration in EENNs. Extensive experiments demonstrate that our approach achieves superior cost-accuracy trade-offs compared to calibration, and EEFP more reliably reflects overall EENN performance. Our code is available at https://github.com/gmum/rethinking-calibration-for-eenns.

## 1. Introduction

The rapid growth of deep learning has led to an increasing demand for resource-efficient models. Early-exit neural networks (EENNs) address this challenge by allowing a model to dynamically process input samples and assign less computation to easy samples, thereby reducing the average inference cost. EENNs enable inference to terminate early by attaching auxiliary classifiers to intermediate layers and halting computation once the prediction is thought to be either correct or sufficiently good as to not make further computation worthwhile. Early-exit mechanisms were originally introduced in the context of vision mod-

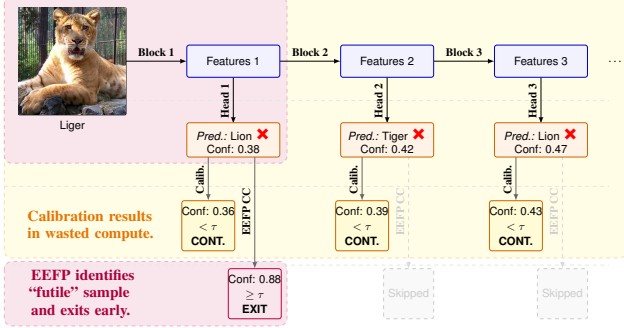

*Figure 1.* Conceptual comparison between standard confidence calibration and our proposed method for early-exit failure prediction. We consider a "futile" sample, which no exit in the model can classify correctly. **Standard Calibration (yellow):** Calibration equates reliability with the probability of correctness, resulting in low confidence (less than exit confidence threshold $\tau$) for samples the model is likely to get wrong. This forces the sample to traverse the entire architecture, causing significant computational waste. **EEFP Confidence Correction (purple):** Our method detects cases for which additional effort is futile, enabling the model to assign high exit confidence and halt execution immediately.

els (Lee et al., 2015; Larsson et al., 2017; Panda et al., 2016; Teerapittayanon et al., 2016; Huang et al., 2018; Kaya et al., 2019), and have since emerged as a natural solution for resource-constrained scenarios (Wang et al., 2019; Tambe et al., 2021; Fang et al., 2020; Ghodrati et al., 2021; Laskaridis et al., 2020; Shao et al., 2021; Li et al., 2020; Jazbec et al., 2023). More recently, early-exit architectures have been successfully adopted in natural language processing (Zhou et al., 2020; Liao et al., 2021; Schuster et al., 2022; Wójcik et al., 2023; Jazbec et al., 2024b), including reasoning models (Yang et al., 2025; Jiang et al., 2025).

The most common early-exit strategy relies on prediction confidence, allowing the network to stop inference when an intermediate classifier produces a prediction whose confidence exceeds a predefined threshold. This introduces a trade-off between prediction quality and computational cost, where higher thresholds generally improve the network accuracy at the expense of increased computation. Naturally, ensuring that exit decisions are correct improves the cost-accuracy trade-off inherent to such dynamic inference systems. As a result, a substantial body of prior work has equated reliable exit decisions with well-calibrated confidence estimates, and in turn, has focused on improving the

---

[1]Jagiellonian University [2]Doctoral School of Exact and Natural Sciences, Jagiellonian University [3]Warsaw University of Technology [4]IDEAS Research Institute [5]Johns Hopkins University. Correspondence to: Bartosz Wójcik <bartwojc@gmail.com>.

*Proceedings of the 43rd International Conference on Machine Learning*, Seoul, South Korea. PMLR 306, 2026. Copyright 2026 by the author(s).

calibration of EENN classifiers under the assumption that better calibration improves overall performance (Qendro et al., 2021; Lilli et al., 2021; Pacheco et al., 2023; Meronen et al., 2024; Mofakhami et al., 2024; Hao et al., 2026).

However, in this work, we demonstrate that this assumption is not entirely correct. We provide a theoretical investigation of calibration in EENNs and show how the motivations for employing EENNs are at odds with the standard notions of calibration. In particular, calibration of individual exits fails to account for the sequential structure of early-exit classifiers. We show that calibration alone—without additional assumptions relating to exit ordering—cannot reliably improve EENN performance.

As an alternative to calibration, we take inspiration from *failure prediction* (Corbière et al., 2019) and propose *Early-Exit Failure Prediction* (EEFP). Crucially, unlike a calibrated exit, an EEFP-aligned exit is *structure aware*: it considers not only the correctness of the current classifier but also the potential correctness of the subsequent classifiers. We introduce a lightweight EEFP-based confidence-correction procedure that serves as an effective alternative to exiting via classifier confidence. Figure 1 conceptually compares our method to confidence-based exiting.

Across a range of standard benchmarks, we demonstrate that networks corrected with our approach consistently achieve better cost–accuracy characteristics than their confidence-calibrated counterparts. Importantly, we also show how our EEFP score reliably captures differences in quality between different EENNs: networks whose classifiers achieve higher EEFP scores perform better, while calibration metrics fail to reflect such differences.

Our contributions are as follows:

- We provide a theoretical analysis showing that standard notions of calibration, developed for isolated classifiers, are insufficient for EENNs.
- We address the limitations of calibration by adapting failure prediction for EENNs and introducing the Early-Exit Failure Prediction (EEFP) score.
- We design an EEFP-aligned confidence-correction procedure that improves cost-accuracy trade-offs in EENNs across multiple benchmarks.

## 2. Background

In this section, we give the necessary background information on early-exit neural networks (EENNs) (2.1) and classifier calibration (2.2).

### 2.1. Early-Exit Neural Networks

EENNs extend standard neural networks, endowing them with intermediate classifiers that perform sequential predictions during inference. They can be represented as a sequence of models $\mathbf{f}^1(\mathbf{x}), \ldots, \mathbf{f}^m(\mathbf{x}), \ldots, \mathbf{f}^M(\mathbf{x})$, each of which maps from a multivariate feature space $\mathbf{x} \in \mathfrak{X}$ to a $(K-1)$-dimensional simplex: $\mathbf{f}^m : \mathfrak{X} \mapsto \Delta^{K-1}$. We assume a $K$-class classification problem with labels taking the form of class indices and denoted $\mathbf{y} \in \mathcal{Y}$. The model's distribution over class labels at step $m$ is then $\mathbf{y} \sim \texttt{Categorical}(\mathbf{f}^m(\mathbf{x}))$, with $\mathrm{f}_y^m(\mathbf{x})$ denoting the classifier's confidence assigned to label $y$. Assume the data is generated from unknown distributions $\mathbf{x} \sim \mathbb{P}(\mathbf{x})$ and $\mathbf{y} \sim \mathbb{P}(\mathbf{y}|\mathbf{x})$, which we can infer only through i.i.d. samples.

Assume that each sub-model of an EENN has an associated computational cost $\lambda_1, \ldots, \lambda_M$ s.t. $\lambda_m \in (0, 1)$, and this overhead is constant across exits (i.e. roughly uniform parameter count between exits): $\lambda_1 = \ldots = \lambda_M$. A very common method for determining when to exit in EENNs is via confidence thresholding: the user must specify a threshold $\tau \in (0, 1)$, and the network will terminate at the $m$-th step if $\max_k f_k^m(\mathbf{x}) > \tau$. The choice of $\tau$ represents the user's preferences for the tradeoff between computation and accuracy: lower $\tau$ reduces the computation at the cost of accuracy, while higher $\tau$ increases the accuracy and compute.

### 2.2. Calibration

Assume for now we are in the simpler case of having only one classifier $\mathbf{f}(\mathbf{x})$, defined just as the EENN component classifiers above. Let $\Phi : \mathfrak{X} \mapsto \mathcal{U}$ be a *grouping function* that partitions the feature space $\mathfrak{X}$ by assigning each point to a group indexed by $u \in \mathcal{U}$. Thus the *equivalence classes* of $\Phi$ are simply the points in feature space that are assigned to the same group: $[\boldsymbol{x}]_\Phi = \{\boldsymbol{x}'|\Phi(\boldsymbol{x}) = \Phi(\boldsymbol{x}')\} \subseteq \mathfrak{X}$. We can now define a general notion of *calibration* (Dawid, 1982):

**Definition 2.1. Calibration**: A predictor $\mathbf{f}$ is (first-order) calibrated w.r.t. a choice of $\Phi$ if

$$\begin{aligned} f_y(\boldsymbol{x}) &= \mathbb{E}\left[\mathbb{P}(\mathbf{y} = y|\mathbf{x}) \mid \mathbf{x} \in [\boldsymbol{x}]_\Phi\right] \\ &= \mathbb{P}\left(\mathbf{y} = y|\Phi(\boldsymbol{x})\right), \ \ \forall \boldsymbol{x} \in \mathfrak{X}, \forall y \in \mathcal{Y}. \end{aligned} \tag{1}$$

This means that $\mathbf{f}$ is calibrated if the confidence it assigns to every $(\boldsymbol{x}, y)$ pair equals the *average* probability of that label *within the group to which $\boldsymbol{x}$ is assigned*. The choice of $\Phi$ naturally gives rise to stronger and weaker forms of calibration (Vaicenavicius et al., 2019), and properly choosing $\Phi$ is known as the *reference class problem* (Hájek, 2007).

The strongest form of calibration is *canonical calibration*, which is when the predictor is actually the Bayes predictor, having matched all the underlying conditional probabilities.

**Definition 2.2. Canonical Calibration**: A predictor $\mathbf{f}$ is *canonically calibrated* if it is calibrated w.r.t. the iden-

tity function $\Phi(\mathbf{x}) = \mathbf{x}$: $f_y(\boldsymbol{x}) = \mathbb{P}(\mathrm{y} = y | \boldsymbol{x})$, $\forall \boldsymbol{x} \in \mathfrak{X}, \forall y \in \mathcal{Y}$.

Canonical calibration is 'strong' because the grouping function is 'fine grained', creating a unique group for every point in feature space. This strength propagates into downstream decision-making: making per-instance decisions by thresholding the predictor (e.g. $f_y(\boldsymbol{x}) \geq 90\%$?) is optimal for controlling Bayes risk since one is thresholding based on the actual conditional probabilities. On the other hand, we can create a weak notion of calibration by considering a 'coarse grained' grouping function. For example, consider a binary classification problem with balanced classes. The prediction $f(\boldsymbol{x}) = 0.5 \; \forall \boldsymbol{x} \in \mathfrak{X}$ is marginally calibrated since it has matched the class base rate. Yet, while the model is calibrated in this weak sense; of course, this is not a good predictive model since it is maximally ambivalent for every feature vector. Here we can also see that making decisions based on thresholding is nonsensical since the model has a constant output: there is no variability in decision making, as the same decision would be taken for all of $\mathfrak{X}$.

In practice, we usually choose a grouping function that is in between the two fine-to-coarse extremes above. In the Machine Learning literature, classifiers are usually evaluated based on *confidence calibration* (Guo et al., 2017), which is when the grouping function is based on the predictive model itself—specifically, by the predictor's maximum confidence over classes.

**Definition 2.3. Confidence Calibration**: Let $\hat{y} = \arg\max_y f_y(\mathbf{x})$. A predictor $\mathbf{f}$ is *confidence calibrated* if it is calibrated w.r.t. $\Phi(\mathbf{x}) = \max_y f_y(\mathbf{x})$:

$$f_{\hat{y}}(\boldsymbol{x}) = \mathbb{P}\left(\mathrm{y} = \hat{y} \; \middle| \; \max_y f_y(\mathbf{x})\right), \quad \forall \boldsymbol{x} \in \mathfrak{X}$$

This choice of group function is rather natural and can be captured with the intuition: *over all the cases for which the predictor is* 70% *confident in the modal class, is the predictor's accuracy* 70%? Confidence calibration is the variant that is of most interest to us. This is because the standard EENN exiting strategy of $\max_k f_k^m(\mathbf{x}) > \tau$, mentioned above, is closely related to the grouping function that defines confidence calibration. A classifier's distance from being calibrated is quantified by *expected calibration error* (ECE). ECE for confidence calibration is computed as

$$\mathrm{ECE}(\mathbf{f}^m) = \mathbb{E}_{\mathbf{x}}\left| \mathbb{P}\left(\mathrm{y} = \hat{y} \; \middle| \; \max_y f_y(\mathbf{x})\right) - f_{\hat{y}}^m(\mathbf{x}) \right|,$$

where again $\hat{y} = \arg\max_y f_y(\mathbf{x})$. Yet in practice, with $f_y^m(\mathbf{x})$ being a floating point number, it is unlikely that there will be groups of sufficient size to estimate $\mathbb{P}(\mathrm{y} = \hat{y} \mid \max_y f_y(\mathbf{x}))$. Thus we must resort to discretizing the interval $[0, 1]$ into 'bins'

(e.g. $[0, .05], (.05, .1], \ldots, (.95, 1.]$), which should be done adaptively to the data at hand (Kumar et al., 2019).

# 3. The Interplay Between Calibration and Adaptive Compute in EENNs

We are now ready to study EENNs through the lens of calibration and investigate how the particular structure and assumptions underlying EENN interact with calibration.

## 3.1. Trivial and Non-Trivial Early-Exit NNs

We begin with a definition of a *useless* EENN.

**Definition 3.1. Trivial EENN**: We call an EENN 'trivial' if all of its constituent predictive models are identical:

$$\mathbf{f}^m(\mathbf{x}) = \mathbf{f}^{m'}(\mathbf{x}), \; \forall m, m' \in [1, M], \; \forall \boldsymbol{x} \in \mathfrak{X}.$$

If all the constituent predictive models return the same probability vector for all inputs, then clearly there is no reason to 'keep around' all $M$ exits. Instead, the modeler should just use $\mathbf{f}^1(\mathbf{x})$ (assuming it has the smallest computational overhead) to make all predictions and discard the $M - 1$ other sub-models. One may also consider 'partially trivial' EENNs, which would have some but not total redundancy in its sub-classifiers. For our theoretical treatment, we assume that all EENNs are either trivial or non-trivial, since any partially trivial EENN can have its redundant sub-models pruned in order to make it non-trivial.

We can now show our first result: that canonically calibrated EENNs are *trivial* EENNs.

**Proposition 3.2. *Canonically Calibrated EENNs are Trivial*:** *If all constituent models of an EENN are canonically calibrated, then the EENN is trivial.*

*Proof*: By the definition of canonical calibration, $\mathbf{f}^m(\mathbf{x}) = \mathbb{P}(\mathrm{y}|\mathbf{x})$, $\forall \boldsymbol{x} \in \mathfrak{X}, \forall m \in [1, M]$. Thus all $M$ sub-models are encoding the same distribution, the Bayes classifier $\mathbb{P}(\mathrm{y}|\mathbf{x})$.

While Proposition 3.2 encodes a best-case scenario that would all but never occur in practice, it shows that for an EENN to be non-trivial, we *expect* model misspecification. Moreover, the sub-models should be misspecified *in unique ways* in order to justify keeping all $M$ exits in the EENN. However, Proposition 3.2 is not necessarily true for other forms of calibration.

**Proposition 3.3. *Calibration Does Not Imply a Trivial EENN*:** *Assume an EENN is calibrated w.r.t. grouping function $\Phi$ for all sub-models. If there exists an equivalence class of size 2 or larger with unique evaluations of $\mathbf{f}^m(\boldsymbol{x})$, then the EENN is not necessarily trivial.*

*Proof*: Assume an EENN is calibrated w.r.t. $\Phi$ *and* is trivial. For an equivalence class containing $\boldsymbol{x}$ and $\boldsymbol{x}'$ such

that $\mathbb{P}(\boldsymbol{x}) = \mathbb{P}(\boldsymbol{x}')$, let the predictive distributions encoded by the $m$th subclassifier be switched, meaning $\mathbf{f}^m(\boldsymbol{x}) \rightarrow \mathbf{f}^m(\boldsymbol{x}')$ and $\mathbf{f}^m(\boldsymbol{x}') \rightarrow \mathbf{f}^m(\boldsymbol{x})$. This new EENN is still calibrated w.r.t. $\Phi$—since permuting $\boldsymbol{f}$'s values within an equivalence class leaves the expectation unchanged—but now $\mathbf{f}^m$ encodes a different predictive model than the other $M-1$ subclassifiers. Thus calibration can be preserved while making the EENN non-trivial. For example, a trivial EENN for a balanced binary prediction task could be marginally calibrated if all exits output $\mathbf{f}^m(\boldsymbol{x}) = .75$ for half of the instances and $\mathbf{f}^m(\boldsymbol{x}') = .25$ for the other half. Exchanging some instances for the $m$th exit to output the opposite value preserves marginal calibration while making the $m$th exit encode a different model from the others.

### 3.2. EENNs with Meaningful Adaptive Computation

Now that we have established that an EENN can be trivial under some forms of calibration (e.g. canonical calibration), but not all, the natural question is: what are properties that we should be looking for in a calibrated EENN? To answer this question, we must first define the concept of *excess conditional risk*:

**Definition 3.4. Excess Conditional Risk**: Assume the log loss. Let $\mathfrak{R}(\mathbf{f}^m, \mathbf{x})$ be the risk under the model at exit $m$, and let $\mathfrak{R}^*(\mathbf{x})$ be the Bayes conditional risk. The excess risk is: $\Delta(\mathbf{f}^m, \mathbf{x}) \triangleq \mathfrak{R}(\mathbf{f}^m, \mathbf{x}) - \mathfrak{R}^*(\mathbf{x})$, with $\Delta(\mathbf{f}^m, \mathbf{x})$ being non-negative and $\mathfrak{R}^*(\mathbf{x}) = \mathbb{H}[\mathbf{y}|\mathbf{x}]$, the irreducible uncertainty in the generative process.

This notion of excess risk is important since, recalling Proposition 3.2, an EENN's exits must be different from the Bayes classifier in order for it to be non-trivial! In other words, there must be excess risk, and we are interested in how it is allocated across the exits. Now with this notion of suboptimality (w.r.t the Bayes classifier), we define the class of ideal EENNs whose excess risk reduces with each exit.

**Definition 3.5. Conditional Anytime EENN**: We say an EENN has a *conditional anytime* property if the excess conditional risk is strictly decreasing with each exit:

$$\Delta(\mathbf{f}^1, \mathbf{x}) > \Delta(\mathbf{f}^2, \mathbf{x}) > \ldots > \Delta(\mathbf{f}^M, \mathbf{x}).$$

We call these 'conditional anytime' EENNs since they have the *anytime* property of using more compute results in lower risk *for each test point* (Zilberstein, 1996; Jazbec et al., 2023), as the EENN is approaching the Bayes classifier with depth. In turn, any subsequent exit is superior to the current one, making the exiting decision entirely a function of one's computational budget. However, most EENNs exhibit only a marginal anytime property (Jazbec et al., 2023), meaning that excess risk decreases not for each point but just on average over $\mathbb{P}(\mathbf{x})$: $\mathbb{E}_{\mathbf{x}}[\Delta(\mathbf{f}^1, \mathbf{x})] > \ldots > \mathbb{E}_{\mathbf{x}}[\Delta(\mathbf{f}^M, \mathbf{x})]$.

Now assume a non-trivial EENN is calibrated with a different grouping function at each exit: $\Phi_1, \ldots, \Phi_M$. The

excess risk at each exit can then be characterized by how informative $\mathbf{x}$ is about $\mathbf{y}$ within the group $[\boldsymbol{x}]_{\Phi_m}$.

**Proposition 3.6. Excess Risk for Calibrated Exits**: *Assume an EENN is non-trivial and calibrated at each exit w.r.t grouping functions $\Phi_1, \ldots, \Phi_M$. Then the EENN's group-conditional excess risk at exit $m \in [1, M]$ is:*

$$\mathbb{E}\left[\, \Delta(\mathbf{f}^m, \mathbf{x}) \mid \mathbf{x} \in [\boldsymbol{x}]_{\Phi_m} \right]$$
$$= \mathbb{H}\left[\mathbf{y}|\Phi_m(\boldsymbol{x})\right] - \mathbb{E}\left[\mathbb{H}\left[\mathbf{y}|\mathbf{x}\right] \mid \mathbf{x} \in [\boldsymbol{x}]_{\Phi_m}\right]$$
$$= \mathcal{I}\left[\, \mathbf{y}, \mathbf{x} \mid \Phi_m(\boldsymbol{x}) \,\right]$$

*where $\mathcal{I}$ denotes mutual information.*

This result shows that a calibrated exit controls excess risk but only on average over $\boldsymbol{x}$'s group. Mutual information decreases as excess risk decreases: $\mathbf{x}$ provides less and less predictive information than what is already encoded in the grouping function. However, this result alone tells us nothing about an EENN having an anytime property. For that to occur, we need to have the grouping functions structured *across* the exits.

**Theorem 3.7. Grouping Refinement Reduces Excess Risk Across Exits**: *If the grouping functions form a refinement chain $\Phi_1 \preceq \cdots \preceq \Phi_M$, then for every pair of exits $(m, e)$ such that $m < e$, we have:*

$$\mathbb{E}[\Delta(\mathbf{f}^m, \mathbf{x}) - \Delta(\mathbf{f}^e, \mathbf{x}) \mid \Phi_m(\boldsymbol{x})] = \mathcal{I}[\mathbf{y}, \Phi_e(\mathbf{x}) \mid \Phi_m(\boldsymbol{x})]$$

*Non-negativity of conditional mutual information then gives a group-conditional anytime property.*

*Proof.* By Proposition 3.6, $\mathbb{E}[\Delta(\mathbf{f}^m, \mathbf{x}) \mid \Phi_m] = \mathcal{I}[\mathbf{y}, \mathbf{x} \mid \Phi_m]$, and analogously $\mathbb{E}[\Delta(\mathbf{f}^e, \mathbf{x}) \mid \Phi_m] = \mathcal{I}[\mathbf{y}, \mathbf{x} \mid \Phi_e, \Phi_m] = \mathcal{I}[\mathbf{y}, \mathbf{x} \mid \Phi_e]$, using $\Phi_m \preceq \Phi_e$ so that $\Phi_e$ already determines $\Phi_m$. The chain rule for conditional mutual information yields $\mathcal{I}[\mathbf{y}, \mathbf{x} \mid \Phi_m] = \mathcal{I}[\mathbf{y}, \Phi_e \mid \Phi_m] + \mathcal{I}[\mathbf{y}, \mathbf{x} \mid \Phi_e, \Phi_m]$, and subtracting gives the claim. $\square$

The key takeaway of this result (and this paper) is that it is not enough for an EENN to be calibrated. Rather, it must have its calibration be well-structured such that later partitions refine earlier ones: $\Phi_1 \preceq \cdots \preceq \Phi_M$. This is the vital property that links computation and calibration.

**Confidence Calibration** Next we consider confidence calibration, a focus of both the machine learning and EENN literatures. We show that it fails to control the excess risk:

**Proposition 3.8. Confidence Calibration Does Not Control Log-Loss-Based Risk**: *Assume $K \geq 3$. For any confidence level $c \in (0, 1)$, there exist confidence-calibrated predictors $\mathbf{f}^m$ with arbitrarily large $\Delta(\mathbf{f}^m, \boldsymbol{x})$ under log-loss.*

*Proof*: Consider any $\boldsymbol{x}$ for which some non-modal class $y' \neq \arg\max_y f_y^m(\boldsymbol{x})$ has positive probability under the

true generative process, $\mathbb{P}(\mathrm{y} = y' \mid \boldsymbol{x}) > 0$. Modify $\mathbf{f}^m$ on this input by setting $f_{y'}^m(\boldsymbol{x}) = \epsilon$ and redistributing the remaining confidence $1 - c - \epsilon$ over the other $K - 2$ non-modal classes. The modal-class probability is unchanged—so the modified predictor is still confidence calibrated—but the pointwise log-loss now contains the term $-\mathbb{P}(\mathrm{y} = y' \mid \boldsymbol{x}) \cdot \log \epsilon$, which diverges as $\epsilon \to 0$. Hence $\mathfrak{R}(\mathbf{f}^m, \boldsymbol{x}) \to \infty$ and $\Delta(\mathbf{f}^m, \boldsymbol{x}) \to \infty$, while confidence calibration is preserved for every $\epsilon$. $\square$ In Proposition 3.6, calibration w.r.t. $\Phi_m$ pins down the entire predictive distribution on each equivalence class, so excess risk is controlled (by mutual information). Confidence calibration, in contrast, controls *only one* of the output coordinates. Since log-loss depends on whichever coordinate matches the true label, the excess risk is not bounded at all. The EENN can therefore fail to satisfy a group-conditional anytime property even though every exit is perfectly confidence calibrated.

However, the failure mode of Proposition 3.8 is specific to log-loss, and if we weaken the excess risk such that it only considers the modal class, then confidence calibration can control the excess risk. Below we consider $\Delta_{\text{0-1}}(\mathbf{f}^m, \mathbf{x})$, the excess risk under the 0-1-loss.

**Lemma 3.9.** *Confidence Calibration Controls 0-1 Excess Risk*: Assuming $\mathbf{f}^m$ is confidence calibrated at level $c \in (0, 1)$, the group-conditional 0-1-excess risk satisfies

$$\mathbb{E}\left[\Delta_{\text{0-1}}(\mathbf{f}^m, \mathbf{x}) \,\middle|\, c = \max_y f_y^m(\boldsymbol{x})\right] =$$
$$\mathbb{E}\left[\max_y \mathbb{P}(\mathrm{y} = y \mid \mathbf{x}) \,\middle|\, c = \max_y f_y^m(\boldsymbol{x})\right] - c.$$

*Proof*: The 0-1 risk at $\boldsymbol{x}$ depends on $\mathbf{f}^m$ only through $\hat{y}(\boldsymbol{x}) = \arg\max_y f_y^m(\boldsymbol{x})$, namely $\mathfrak{R}_{\text{0-1}}(\mathbf{f}^m, \boldsymbol{x}) = \mathbb{P}(\mathrm{y} \neq \hat{y}(\boldsymbol{x}) \mid \boldsymbol{x})$. Averaging over the equivalence class of $c$ and applying confidence calibration, $\mathbb{P}(\mathrm{y} = \hat{y}(\mathbf{x}) \mid c) = c$, gives $\mathbb{E}[\mathfrak{R}_{\text{0-1}}(\mathbf{f}^m, \mathbf{x}) \mid c] = 1 - c$. The Bayes 0-1 risk averaged over the class is $\mathbb{E}[1 - \max_y \mathbb{P}(\mathrm{y} = y \mid \mathbf{x}) \mid c]$, and subtracting yields the stated excess. $\square$ Even though confidence calibration can control excess risk under an appropriate (matching) choice of loss function, we still need the additional, structural ordering of the grouping function in order for the EENN to support anytime computation.

# 4. Early Exit Failure Prediction (EEFP)

As argued in the previous section, classifier calibration that is localized per-exit is ill-suited to EENNs, as its underlying assumptions are fundamentally misaligned with EENN design. Calibration, without inter-exit structural assumptions, ignores the computational cost induced by subsequent exit decisions. Therefore, we next consider an alternative prediction problem that explicitly accounts for the utility of subsequent classifiers in the exit decision. We show that

when this alternative meta-model is calibrated, it contains the necessary awareness of the inter-exit quality.

## 4.1. Failure Prediction for EENNs

We begin by formulating the ideal exiting criteria for EENNs. Given a true label y and prediction $y_m$ of the $m$-th classifier in the EENN, the optimal stopping decision for exit $m$, denoted $\otimes_m$, is:

$$\otimes_m = \begin{cases} 1, & \text{if } y_m = \mathrm{y} \ \lor \ \forall_{e \geq m} \ y_e \neq \mathrm{y} \\ 0, & \text{otherwise.} \end{cases} \quad (2)$$

This means that the EENN should exit at the $m$-th classifier ($\otimes_m = 1$) if its prediction is correct *or* if none of the subsequent classifiers would be correct. Consequently, $\otimes_m = 1$ indicates that exiting at depth $m$ is predictively *and* computationally optimal: processing the sample with deeper classifiers cannot produce a better prediction with less computation. We refer to this task as *failure prediction* (Corbière et al., 2019; Hendrycks & Gimpel, 2016) to emphasize its awareness of computation.

Failure prediction is challenging as it must be done without access to the true label *and* without obtaining the predictions of future exits. Yet this prediction task should be tractable in many cases, as it does not require $p(\otimes_m|\mathbf{x})$ to know exactly what label the future exits will predict. Rather, it only requires inferring the binary variable of whether *some* unspecified future exit(s) will be correct. A similar assumption is employed by the *learning to defer* literature (Mozannar & Sontag, 2020; Verma et al., 2023) for modeling human prediction quality without having to fully encode the human's behavior within a model.

**Calibration** Given a fixed EENN and prediction task, the underlying generative model for failure prediction is:

$$\mathbb{P}(\otimes_m = 1|\mathbf{x}) = \mathbb{P}(\mathrm{y}_m = \mathrm{y}|\mathbf{x}) + \prod_{e=m}^{M} \mathbb{P}(\mathrm{y}_e \neq \mathrm{y}|\mathbf{x})$$
$$\text{where} \quad \mathbb{P}(\mathrm{y}_m = \mathrm{y}|\mathbf{x}) = \sum_{y' \in \mathcal{Y}} \mathbb{P}(\mathrm{y} = y'|\mathbf{x}) \cdot f_{y'}^m(\mathbf{x})$$

and $\mathbb{P}(\mathrm{y}_m \neq \mathrm{y}|\mathbf{x}) = 1 - \mathbb{P}(\mathrm{y}_m = \mathrm{y}|\mathbf{x})$. Thus $\mathbb{P}(\mathrm{y}_m = \mathrm{y}|\mathbf{x})$ denotes the probability of the predictions sampled from $\boldsymbol{f}^m$ colliding with the true label sampled from $\mathbb{P}(\mathrm{y}|\mathbf{x})$. Now assume the EENN is calibrated w.r.t. grouping functions $\Phi_1, \ldots, \Phi_M$. The next result links the anytime property to a monotonically-increasing probability of stopping:

**Proposition 4.1.** *Refinement Implies Monotone Stopping:* Assume the setting of group-conditional anytime computation: each exit $\mathbf{f}^m$ is calibrated w.r.t $\Phi_m$ and the partitions are ordered by refinement $\Phi_1 \preceq \cdots \preceq \Phi_M$. Then the EEFP-style stopping probability is non-decreasing in $m$, in

*expectation over each equivalence class of the coarser exit:*

$$\mathbb{E}\big[\,\mathbb{P}(\otimes_m = 1 \mid \mathbf{x})\,\big|\,[\boldsymbol{x}]_{\Phi_m}\big] \leq$$
$$\mathbb{E}\big[\,\mathbb{P}(\otimes_{m+1} = 1 \mid \mathbf{x})\,\big|\,[\boldsymbol{x}]_{\Phi_m}\big] \quad \forall m \in [1, M{-}1].$$

The proof is provided in Appendix A. While failure prediction is sensitive to the partition-granularity of the current exit vs future ones, its view of future exits is in aggregate. In other words, the probability of stopping is the same regardless of whether the correct prediction is expected to arrive at the next exit or the very last one. To achieve this level of sensitivity, one must modify the product $\prod_{e=m}^{M} \mathbb{P}(\mathrm{y}_e \neq \mathrm{y}|\mathbf{x})$ so that it is not permutation invariant. We leave this as a topic for future work.

### 4.2. Meta-Classifier for Failure Prediction

We propose a post-training method for EENNs that directly predicts the optimal stopping decision $\otimes_m$. Instead of relying on standard calibration methods such as temperature calibration, which ignore downstream computation, we train lightweight meta-models to estimate the probability of stopping at a given depth. Based on the predictions of the current classifier, these modules model the optimal stopping decision $\otimes_m$ by outputting a scalar stopping probability. This approach therefore provides an alternative confidence score that explicitly accounts for inter-exit dynamics.

Our first implementation operates using the exit classifier's confidences as inputs:

$$p\left(\otimes_{n,m} = 1|\mathbf{x}_n; \mathbf{f}^m\right) \triangleq g^m\left(f^m_{\pi_1}(\mathbf{x}_n), \ldots, f^m_{\pi_k}(\mathbf{x}_n)\right) \quad (3)$$

where $\otimes_{n,m}$ is the stopping decision of the $m$-th exit for the $n$-th sample, and $\pi_1, \ldots, \pi_k$ denotes the top-$k$-ranked (e.g. $k = 5$) class indices according to a descending ordering of $\mathbf{f}^m$'s confidences. In our experiments, $g^m$ is implemented as a multi-layer perceptron (MLP).

Our second implementation leverages the predictions made at previous exits as additional input features:

$$p\left(\otimes_{n,m} = 1|\mathbf{x}_n; \mathbf{f}^1, \ldots, \mathbf{f}^m\right) \triangleq$$
$$g^m_h\left(\boldsymbol{f}^1_{\boldsymbol{\pi}(m,k)}(\mathbf{x}_n), \ldots, \boldsymbol{f}^m_{\boldsymbol{\pi}(m,k)}(\mathbf{x}_n)\right) \quad (4)$$

where $\boldsymbol{\pi}(m,k)$ denotes the top-$k$-ranked class indices as computed for the $m$-th exit's classifier confidences. We give this model the subscript $h$ for 'history', as it has the ability to see how the top-$k$-classes' confidences evolved over the exits that have been evaluated so far. Again $g^m_h$ is implemented as an MLP whose architecture is the same except for the change in feature dimensionality.

To train both implementations, we first train the EENN of interest and freeze its weights. Then, for each exit $m \in [1, M]$, we train the corresponding failure prediction model $g$ using the binary cross-entropy loss on a held-out set. During training, predictions from all $M$ classifiers are collected to compute true stopping targets, as defined in Eq. (2). Once trained, $g$'s output confidences can be used for threshold-based exiting, just as the classifier confidences are typically used: if $p(\otimes_m = 1|\mathbf{x}) > \tau$, then exit.

### 4.3. Metric for Failure Prediction

To measure the ability to predict failure, we propose an EEFP score for $m$-th internal classifier using the observed value of $\otimes_m$, as defined in Equation (2), and confidence scores $c_{n,m}$ (i.e. the $m$-th exit's score for the $n$-th sample):

$$\mathrm{EEFP}\Big(m; \{\boldsymbol{x}_n, y_n, y_{n,m}, \ldots, y_{n,M}\}_{n=1}^N\Big) \triangleq$$
$$\mathrm{AUROC}\Big(\{c_{n,m}, \otimes_{n,m}\}_{n=1}^N\Big) \quad (5)$$

where $y_n$ denotes the true label for the $n$-th sample and $y_{n,m} = \arg\max_y f^m_y(\boldsymbol{x}_n)$ the prediction of the $m$-th exit for the $n$-th sample. The confidence scores $c$ are computed in two ways, to compare EEFP with traditional classifier-confidence-based exiting. For the former, $c_{n,m} = g^m(\cdot)$. For the latter, $c_{n,m} = \max_y f^m_y(\mathbf{x}_n)$. AUROC measures the probability that a randomly chosen correct prediction receives a higher confidence score than a randomly chosen incorrect one, reflecting how well the confidence scores separate the true stopping decisions.

## 5. Experiments

We conduct experiments in standard computer vision settings and evaluate early-exit ResNet-34 (He et al., 2016), ViT-Tiny, ViT-Small (Dosovitskiy et al., 2021), Efficient-Net (Tan et al., 2019), and MSDNet (Huang et al., 2018) on CIFAR100 (Krizhevsky & Hinton, 2009), TinyImageNet (Yang, 2015), and ImageNet-1k (Deng et al., 2009) datasets. We train baseline early-exit models from scratch, with the exception of the ImageNet models, for which we start from a pretrained checkpoint. We then freeze the weights of the resulting models and use them as the starting point for two post-training procedures: temperature calibration (described in detail in Appendix B and Appendix C) and an EEFP-inspired confidence correction method (described in detail in Appendix C). Unless stated otherwise, we use the "history" variant of confidence correction defined in Equation (4), with k = 5 in the top-$k$ operator as our primary method in all the analyzed settings.

We evaluate the methods by considering the compute–accuracy trade-off of the whole EENN, as well as the expected calibration error (ECE) and our proposed EEFP score obtained for the individual classifiers. To obtain the compute–accuracy characteristics, we dynamically evaluate the networks with varying exit thresholds and plot the average compute per sample (measured in FLOPs) and the

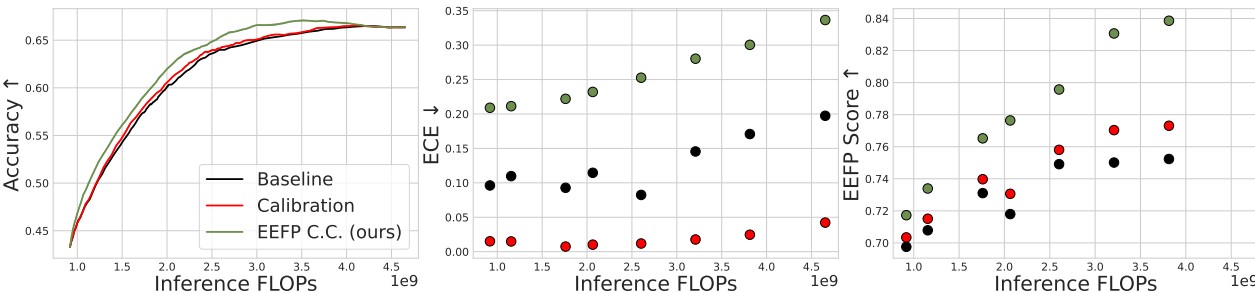

*Figure 2.* Cost-accuracy trade-off, alongside the expected calibration error (ECE) and our proposed early-exit failure prediction (EEFP) score for each intermediate classifier, for Resnet-34 on the TinyImageNet. We compare the standard early-exit model, temperature-calibrated one and one with our proposed EEFP-motivated confidence correction procedure. The results show that **our procedure achieves the best cost-accuracy trade-off**, and that calibration is insufficient to measure the quality of early-exit classifiers: **higher EEFP score correctly reflects the quality of early exit models, while networks with lower ECE actually exhibit worse performance**.

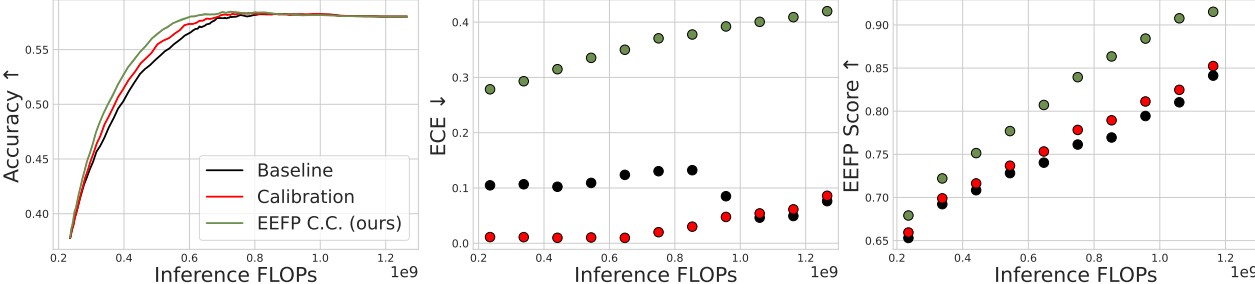

*Figure 3.* Cost-accuracy, ECE and EEFP for ViT-Tiny on TinyImageNet (as in Figure 2). Despite changing the model architecture, the experimental results point to the same conclusions: our proposed EEFP score is way more reliable than calibration for analyzing EENNs.

classification accuracy corresponding to the thresholds. This enables us to analyze how each metric correlates with the end-to-end performance of EENNs and to assess their suitability as indicators of overall early-exit behavior.

### 5.1. Standard Benchmarks

To evaluate the effectiveness of our confidence correction and the usefulness of EEFP for comparing EENNs, we consider three variants of EENNs: baseline networks, temperature-calibrated ones, and EENNs enhanced with our confidence correction. Specifically, we train EENNs on top of ResNet-34 and ViT-Tiny architectures on TinyImageNet, and show the obtained results in Figures 2 and 3.

The results are consistent across both architectures, and the networks with our proposed correction obtain clearly better cost-accuracy characteristics, although calibrated networks also achieve slight improvements over the baseline. Temperature calibration reduces ECE and slightly improves the EEFP score. In contrast, our confidence correction increases ECE while improving EEFP. Importantly, the relative quality of the models is accurately reflected by the EEFP scores of the internal classifiers, whereas ECE does not reliably predict end-to-end early-exit performance, supporting the validity of our proposed metric and approach.

We present additional results for MSDNet on CI-FAR100, EfficientNet-B2 on ImageNet-1k and ViT-Small on ImageNet-1k in Table 1 (standard deviations are reported in Table 7), reporting averaged accuracies at selected compute budgets alongside the EEFP score and ECE averaged across internal classifiers. Consistent with earlier experiments, our confidence correction method achieves superior compute–accuracy trade-offs, and EEFP again provides a reliable prediction of model performance.

### 5.2. Impact of the Top-$k$ Selection

Our confidence correction procedure (Section 4.2) applies a top-$k$ operator over class predictions to reduce computational overhead. Therefore, the choice of $k$ naturally affects the performance of our method, with smaller $k$ reducing computational overhead, but skipping information potentially relevant to learning a reliable correction function. To assess the impact of $k$ and identify its most effective value, we perform an ablation study on ResNet34 trained on Tiny-ImageNet and report the accuracy at selected FLOPs budgets and the average EEFP score across all classifiers in Table 2 (standard deviations are reported in Table 9).

Our approach achieves robust performance across different $k$ values. Interestingly, an EENN with a smaller $k$ even slightly outperforms a variant which uses all predictions ($k = 200$), likely due to the reduced complexity of the

*Table 1.* Accuracy for selected FLOPs budgets, ECE, and EEFP scores over the classifiers for ImageNet experiments. Consistently with our previous results, EEFP score better correlates with the actual network performance, and our confidence correction procedure provides a better alternative to standard calibration.

| | FLOPs threshold | | | ECE($\Downarrow$) | EEFP($\Uparrow$) |
|---|---|---|---|---|---|
| | 25% | 50% | 75% | | |
| *MSDNet - CIFAR-100* | | | | | |
| Baseline | 71.38 | 75.01 | 75.48 | 0.21 | 0.82 |
| Calibrated | 71.92 | 75.30 | 75.46 | **0.02** | 0.83 |
| Ours | **73.27** | **75.66** | 75.58 | 0.15 | **0.86** |
| *ViT Small - ImageNet1000* | | | | | |
| Baseline | 39.67 | 71.03 | 80.19 | 0.16 | 0.81 |
| Calibrated | 39.40 | 72.04 | 80.34 | **0.02** | 0.82 |
| Ours | **40.18** | **72.45** | **80.41** | 0.15 | **0.84** |
| *EfficientNet B2 - ImageNet1000* | | | | | |
| Baseline | 33.99 | 62.16 | 76.88 | 0.18 | 0.74 |
| Calibrated | 33.70 | 62.99 | 77.49 | **0.02** | **0.76** |
| Ours | **34.50** | **63.46** | **77.57** | 0.15 | **0.76** |

*Table 2.* Accuracy and EEFP score depending on the top-$k$ in our confidence correction for ResNet34 evaluated on TinyImagenet.

| | FLOPs threshold | | | EEFP($\Uparrow$) |
|---|---|---|---|---|
| | 25% | 50% | 75% | |
| top-1 | 48.81 | 62.82 | 66.45 | 0.74 |
| top-2 | 49.86 | 63.79 | 66.94 | **0.78** |
| top-5 | **50.67** | **64.06** | **67.03** | **0.78** |
| top-10 | **50.67** | 63.91 | 66.86 | **0.78** |
| top-200 | 50.57 | 63.36 | 66.39 | 0.76 |

correction module leading to a regularization effect, and a slightly lower FLOPs per threshold. This effect can be further explained by the closer investigation of the inputs to the correction module: since the module operates on probability distributions produced by softmax, which sum to 1, only a few of the entries in the probability distribution are meaningfully larger than 0. Consequently, increasing $k$ beyond the first few entries feeds the correction module with useless values from the distribution tail, which may even introduce noise. Consistently with the previous experiments, higher EEFP scores in this ablation also continue to correspond to better cost-accuracy trade-offs. Overall, these results support the use of the top-$k$ operator in our correction approach.

### 5.3. Confidence Correction with Previous Predictions

We propose two variants of our confidence correction procedure, $g$ and $g_h$, defined in Equation (3) and Equation (4) re-

spectively. The key difference between the two approaches is that the first one uses only the predictions of the corresponding classifier as input, while the second additionally leverages information specific to early-exit networks: the history of all previous predictions up to the current classifier. To evaluate the impact of incorporating this historical information, we perform an ablation study of these two approaches, and compare the results in Table 3.

*Table 3.* Accuracy and EEFP score depending on the use of previous predictions in our confidence correction (C.C.) procedure for ResNet34 evaluated on TinyImagenet, compared with the baseline model and standard temperature-based calibration.

| | FLOPs threshold | | | EEFP($\Uparrow$) |
|---|---|---|---|---|
| | 25% | 50% | 75% | |
| Baseline | 48.20 | 62.44 | 65.66 | 0.73 |
| Calibration | 48.91 | 62.98 | 65.73 | 0.74 |
| W/o History | **50.82** | 63.42 | 66.23 | 0.77 |
| With History | 50.67 | **64.06** | **67.03** | **0.78** |

Confidence correction with history improves accuracy at middle-to-high compute budgets, where information from previous classifiers can be effectively leveraged. At lower compute budgets, the simpler, historyless correction performs slightly better, likely because early classifiers are weak and their predictions are less informative. Nevertheless, the historyless variant still outperforms standard calibration. The compute-accuracy behavior of the compared models is consistently well captured by the EEFP score, as observed in the previous experiments.

### 5.4. Failure Cases of Calibrated EENNs

Finally, to further demonstrate the usefulness of the EEFP score, we highlight empirical cases where deliberately miscalibrated EENNs outperform the calibrated ones, and where standard calibration metrics such as ECE fail to correctly rank model performance in contrast to our proposed EEFP score. We calibrate the intermediate classifiers of MSDNet models trained on CIFAR100 using temperature scaling (Guo et al., 2017), and subsequently create two deliberately decalibrated variants by multiplying the temperature in the intermediate classifiers by 3.0 or 0.3, resulting in underconfident and overconfident models, respectively. We then evaluate cost–accuracy curves, the expected calibration error (ECE) of the intermediate classifiers, and their EEFP scores and show the results in Figure 4. Overconfident models achieve a more favorable cost–accuracy trade-off despite substantially worse calibration scores; their performance, however, is accurately captured by the EEFP metric.

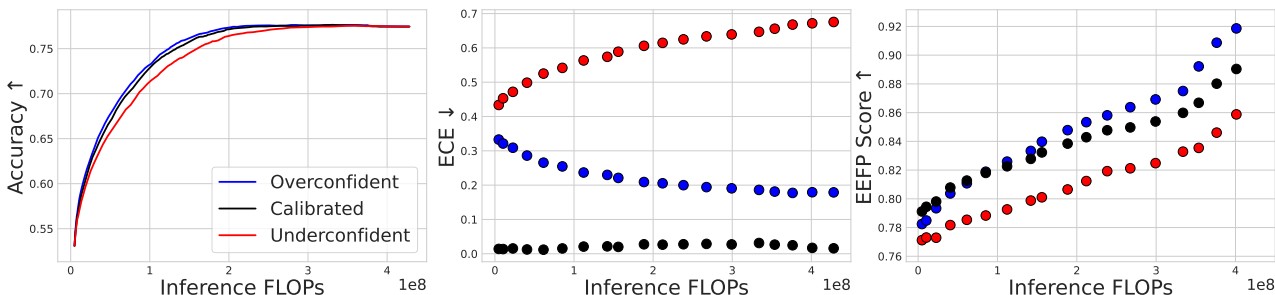

*Figure 4.* Cost-accuracy, ECE and EEFP for MSDNet on the CIFAR-100 for one calibrated model and two decalibrated models with modified temperature values. Calibration fails to capture the quality of the early-exit model, as an overconfident network with higher ECEs performs better than the calibrated one. On the other hand, our proposed EEFP score accurately reflects the quality of the EENNs.

## 5.5. Computational Cost of Confidence Correction

One can easily estimate the cost of a single history variant confidence corrector. In our experiments, each corrector is a 2-layer MLP with hidden dim equal to 128. The size of the input for the $n$-th corrector is $nk$. Since we use $k = 5$ for top-$k$ by default, this gives $640n + 128$ as an estimate of MAC operations, e.g. 6528 for the 10-th exit. In Table 4 we report the FLOPs as measured for the early-exit ResNet-34 models executed on 64x64 inputs. The computational cost of our confidence correctors is negligible.

*Table 4.* FLOPs of a standard EENN and an EENN with our confidence correction. The confidence correction increases the computational cost of the main model by approximately 0.001%.

| Model | FLOPs |
|---|---|
| Baseline EENN | 4 653 372 423 |
| Ours | 4 653 433 635 |

## 6. Related Work

**Early-Exit Neural Networks** Several works on EENNs have examined the calibration of intermediate classifiers as a way to improve performance (Pacheco et al., 2023; Mofakhami et al., 2024; Wójcik et al., 2023). Yet calibration failures have motivated alternative approaches for uncertainty quantification and consistency of the exiting policy, such as training a gating mechanism (Regol et al., 2024), post-hoc re-calibration for monotonicity (Jazbec et al., 2023), anytime-valid hypothesis testing (Jazbec et al., 2024a), distribution-free risk control (Schuster et al., 2022; Jazbec et al., 2024b), and Bayesian treatments of the classifiers (Meronen et al., 2024).

**Model Cascades** Our analysis of calibration and EENNs equally applies to model cascades (Viola & Jones, 2001; Saberian & Vasconcelos, 2014; Marquez et al., 2018). Cascades, too, can be formulated as a sequence of models for which we expect later models to outperform earlier ones (to justify the additional computation). Thus, confidence calibration alone is not sufficient for a well-adaptive cascade,

as has been pointed out by Jitkrittum et al. (2023) and Regol et al. (2025). However, our proposed remedy of failure prediction does not translate as well to cascades since the models in a cascade often have sizes that jump in orders of magnitude as the sequence progresses. In order to predict if the subsequent models will fail, the failure prediction model itself will need to be quite powerful.

## 7. Conclusion, Limitations, and Future Work

In this work, we challenge the assumption that confidence calibration of an EENN's intermediate classifiers improves performance. We highlight that calibration does not innately account for computational cost. To address these limitations, we propose an approach based on failure prediction (EEFP). Critically, EEFP considers both prediction correctness and the cost of continuing inference—factors ignored by traditional formulations of calibration. We propose a lightweight meta-classifier that models the stopping probability under failure prediction and show that it consistently improves the efficiency of EENNs across diverse benchmarks.

The central limitation of our work is that we do not propose a method for better calibrating an EENN by imposing the structural ordering of the per-exit grouping functions required for anytime computation (Theorem 3.7). Furthermore, the grouping functions Φ used in our theoretical analysis remained abstract and measuring their granularity, in practice, would be challenging and nearly impossible for high-dimensional problems. Yet, we hope this work will improve the field's understanding of EENNs and inspire future research that guarantees an EENN will have the inter-exit structure that enables adaptive computation and early recognition of hopelessly difficult instances. Moreover, we believe our insights can also be applied to adaptive computation with large langauge models, such as in chain-of-thought reasoning (Wei et al., 2022; Wang et al., 2026).

## Acknowledgements

We thank Metod Jazbec for helpful discussions. Filip Szatkowski was funded by National Science Centre (NCN, Poland) Grant No. 2022/45/B/ST6/02817.

## Impact Statement

The goal of our paper is to advance the theoretical understanding and efficiency of early-exit neural networks. We aim for our research to contribute to the development of adaptive computation models that reduce overall computational cost, enabling broader deployment of neural networks in resource-constrained settings. Our methods are broadly applicable to neural network models and, more generally, to dynamic computation machine learning systems. While more efficient neural networks may have diverse societal consequences, we do not identify any specific impact that warrants highlighting here; we consider potential risks and impacts to be specific to particular applications of neural networks within subfields of computer science.

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

# Appendix

## A. Proof of Proposition 4.1

*Proof.* Decompose

$$\mathbb{P}(\otimes_m = 1 \mid \boldsymbol{x}) = \mathbb{P}(\mathrm{y}_m = \mathrm{y}|\mathbf{x}) + \prod_{e=m}^{M} \mathbb{P}(\mathrm{y}_e \neq \mathrm{y}|\mathbf{x}) = A_m(\boldsymbol{x}) + B_m(\boldsymbol{x})$$

with $A_m$ being the collision probability and $B_m$ being the "all remaining exits fail" product.

*Step 1 (the B term, pointwise).* Since each factor lies in $[0,1]$, $B_{m+1}(\boldsymbol{x}) = B_m(\boldsymbol{x}) / \mathbb{P}(y_m \neq \mathrm{y} \mid \boldsymbol{x}) \geq B_m(\boldsymbol{x})$ for every $\boldsymbol{x}$, so $\mathbb{E}[B_{m+1} \mid [\boldsymbol{x}]_{\Phi_m}] \geq \mathbb{E}[B_m \mid [\boldsymbol{x}]_{\Phi_m}]$.

*Step 2 (the A term, via refinement and Jensen).* By calibration of $\mathbf{f}^m$, $f_{y'}^m(\boldsymbol{x}) = \mathbb{P}(\mathrm{y} = y' \mid \Phi_m(\boldsymbol{x}))$, so the tower property gives $\mathbb{E}[A_m \mid \Phi_m] = \sum_{y'} \mathbb{P}(\mathrm{y} = y' \mid \Phi_m)^2$. The same calculation at exit $m+1$, conditioned down to $\Phi_m$ using $\Phi_m \preceq \Phi_{m+1}$, yields $\mathbb{E}[A_{m+1} \mid \Phi_m] = \sum_{y'} \mathbb{E}[\mathbb{P}(\mathrm{y} = y' \mid \Phi_{m+1})^2 \mid \Phi_m] \geq \sum_{y'} \mathbb{P}(\mathrm{y} = y' \mid \Phi_m)^2 = \mathbb{E}[A_m \mid \Phi_m]$, where the inequality is Jensen on the convex map $p \mapsto p^2$.

Adding Steps 1 and 2 proves the claim. $\qquad\square$

## B. Influence of Temperature Scaling

### B.1. Probability distribution after temperature scaling

In a classification problem with $K$ classes, suppose the model outputs logits $z_1, \ldots, z_K$ for a given data sample, and let $r$ denote the index of the most probable class. Without temperature scaling, the softmax probabilities are

$$p_k = \frac{e^{z_k}}{\sum_{l=1}^{C} e^{z_l}}, \quad k = 1, \ldots, K.$$

By introducing

$$d := \log\left(\sum_{l=1}^{C} e^{z_l}\right),$$

we can equivalently write

$$p_k = e^{z_k - d}, \quad \text{and hence} \quad z_k = \log(p_k) + d.$$

When scaling logits by a temperature parameter $T > 0$, the probabilities become

$$p_k^{(T)} = \frac{e^{z_k/T}}{\sum_{l=1}^{C} e^{z_l/T}} = \frac{p_k^{1/T}}{\sum_{l=1}^{C} p_l^{1/T}}, \quad k = 1, \ldots, C.$$

Therefore, the confidence changes from $p_r$ to

$$p_r^{(T)} = \frac{p_r^{1/T}}{\sum_{l=1}^{C} p_l^{1/T}}.$$

Since the denominator $\sum_{l=1}^{C} p_l^{1/T}$ depends on the entire probability distribution (and not only on $p_r$), two samples with the same original confidence $p_r$ can yield different scaled confidences $p_r^{(T)}$ after temperature scaling.

### B.2. Temperature scaling does not preserve the ranking of samples

Consider a toy example with four classes. The $j-th$ classifier's logit outputs are shown in Table 5, while the corresponding softmax probabilities are reported in Table 6. Confidences of the predictions are as follows: $c_j(A) \approx 0.450$, $c_j(B) \approx 0.400$,

*Table 5.* Classifier's logits in a toy problem.

|          | Class 1 | Class 2 | Class 3 | Class 4 |
|----------|---------|---------|---------|---------|
| Image $A$ | -0.7985 | -0.9163 | -2.3026 | -2.9957 |
| Image $B$ | -1.6094 | -1.6094 | -0.9163 | -1.6094 |
| Image $C$ | -1.2040 | -0.9676 | -1.3471 | -2.8134 |

*Table 6.* Classes probability distribution (after softmax) in a toy problem.

|          | Class 1 | Class 2 | Class 3 | Class 4 |
|----------|---------|---------|---------|---------|
| Image $A$ | 0.450 | 0.400 | 0.100 | 0.050 |
| Image $B$ | 0.200 | 0.200 | 0.400 | 0.200 |
| Image $C$ | 0.300 | 0.380 | 0.260 | 0.060 |

$c_j(C) \approx 0.380$. Therefore the ranking of samples is $A, B, C$ (from the most to the least confident ones). However, when temperature changes, the ranking does as well. For example for temperature 0.3, $c_j(A) \approx 0.594$, $c_j(B) \approx 0.771$, $c_j(C) \approx 0.575$, and the ranking changes to $B, A, C$. On the other hand, for temperature 3.0, $c_j(A) \approx 0.328$, $c_j(B) \approx 0.296$, $c_j(C) \approx 0.299$, and the ranking changes to $A, C, B$.

## C. Experimental Setup Details

This section describes the experimental setup used throughout the experiments reported in the main paper.

### C.1. Training

All models are trained using the AdamW optimizer (Loshchilov & Hutter, 2019). We employ a cosine annealing learning rate scheduler with warm restarts and a linear warm-up phase. Data augmentation includes random resizing, cropping, rotation, contrast adjustment, random erasing, Mixup (Zhang et al., 2018), and CutMix (Yun et al., 2019). Training is performed until convergence using an early-stopping criterion, with a maximum limit of 1000 epochs.

### C.2. Calibration

The calibration of the early-exit network is performed using a gradient-based approach on a held-out validation set that is not used to optimize the EENN. During calibration, the EENN parameters are frozen, and temperature scaling modules are attached to each exit head. The calibration objective minimizes the negative log-likelihood (NLL). Each exit head is calibrated independently with its own temperature parameter. For each experimental configuration, three calibration models are trained, each with different initial random weights. No data augmentations are applied during calibration in order to match the test-time data distribution as closely as possible.

### C.3. Confidence Correction

EEFP Confidence Correctors (C.C.) consist of two-layer MLPs with a hidden dimension of 128 and are trained on the same held-out dataset used for calibration. The EENN parameters remain frozen, and each C.C. head is optimized independently by minimizing the binary cross-entropy loss between the target confidence defined in Equation (2) and the predicted value. Each C.C. head has its own set of learnable parameters. As in calibration, three C.C. models are trained per experiment, each with different initial random weights. Confidence corrector training is performed without data augmentation to ensure consistency with the test-time data distribution.

### C.4. Evaluation in the Early-Exit Environment

For each model and each exit head, the decision thresholds are determined using a validation set that is disjoint from the test set. Threshold selection follows the heuristic proposed by (Huang et al., 2018), which derives exit-specific thresholds based on the empirical distribution of confidence scores.

Given a predefined FLOPs budget, the network is expected to terminate at each exit for a prescribed fraction of input samples. The allocation of samples across exits is controlled by a parameter $q$. Specifically, at the $j$-th exit, the required fraction of samples that terminate is defined as:

$$\text{exit-share}(j) = \frac{q^j}{\sum_{l=0}^{J-1} q^l},$$

During inference on the test set, each model computes confidence scores for incoming samples and applies its corresponding set of thresholds to determine the exit point and, consequently, the final prediction.

## D. Full Results

Due to space constraints, we presented only the averaged results in the main article. In Table 7 and Table 8 we present the full results including standard deviations for Table 1 and Table 3, respectively.

*Table 7.* Accuracy, ECE, and EEFP scores for selected FLOPs budgets for basline EENN, calibrated EENN and after our EEFP confidence correction method.

| | FLOPs threshold | | | ECE($\Downarrow$) | EEFP($\Uparrow$) |
|---|---|---|---|---|---|
| | 25% | 50% | 75% | | |
| MSDNet - CIFAR-100 | | | | | |
| Baseline | 71.38 | 75.01 | 75.48 | 0.21 | 0.82 |
| Calibrated | $71.92 \pm 0.00$ | $75.30 \pm 0.00$ | $75.46 \pm 0.00$ | $\mathbf{0.02 \pm 0.00}$ | $0.83 \pm 0.00$ |
| Ours | $\mathbf{73.27 \pm 0.05}$ | $\mathbf{75.66 \pm 0.05}$ | $\mathbf{75.58 \pm 0.03}$ | $0.15 \pm 0.00$ | $\mathbf{0.86 \pm 0.00}$ |
| ViT Small - ImageNet1000 | | | | | |
| Baseline EE | 39.67 | 71.03 | 80.19 | 0.16 | 0.81 |
| Calibrated | $39.40 \pm 0.00$ | $72.04 \pm 0.00$ | $80.34 \pm 0.00$ | $\mathbf{0.02 \pm 0.00}$ | $0.82 \pm 0.00$ |
| Ours | $\mathbf{40.18 \pm 0.01}$ | $\mathbf{72.45 \pm 0.04}$ | $\mathbf{80.41 \pm 0.01}$ | $0.15 \pm 0.00$ | $\mathbf{0.84 \pm 0.00}$ |
| EfficientNet B2 - ImageNet1000 | | | | | |
| Baseline EE | 33.99 | 62.16 | 76.88 | 0.18 | 0.74 |
| Calibrated | $33.70 \pm 0.01$ | $62.99 \pm 0.01$ | $77.49 \pm 0.00$ | $\mathbf{0.02 \pm 0.00}$ | $\mathbf{0.76 \pm 0.00}$ |
| Ours | $\mathbf{34.50 \pm 0.04}$ | $\mathbf{63.46 \pm 0.03}$ | $\mathbf{77.57 \pm 0.01}$ | $0.15 \pm 0.00$ | $\mathbf{0.76 \pm 0.00}$ |

*Table 8.* Accuracy and EEFP scores of two variants of our EEFP confidence correction procedure for ResNet34 evaluated on TinyImagenet, compared with the baseline model and temperature scaling calibration.

| | FLOPs threshold | | | ECE($\Downarrow$) | EEFP($\Uparrow$) |
|---|---|---|---|---|---|
| | 25% | 50% | 75% | | |
| Baseline | 48.20 | 62.44 | 65.66 | 0.13 | 0.73 |
| Calibrated | $48.91 \pm 0.01$ | $62.98 \pm 0.01$ | $65.73 \pm 0.00$ | $\mathbf{0.02 \pm 0.00}$ | $0.74 \pm 0.00$ |
| W/o History | $\mathbf{50.82 \pm 0.03}$ | $63.42 \pm 0.07$ | $66.23 \pm 0.01$ | $0.25 \pm 0.00$ | $0.77 \pm 0.00$ |
| With History | $50.67 \pm 0.01$ | $\mathbf{64.06 \pm 0.06}$ | $\mathbf{67.03 \pm 0.03}$ | $0.26 \pm 0.00$ | $\mathbf{0.78 \pm 0.00}$ |

# E. Extended Comparison to Calibration Methods

In the main paper, we compared our method only to temperature scaling, which is one of the most popular post-hoc calibration methods. In Table 9 we provide extended evaluation results that also include vector scaling and matrix scaling (Guo et al., 2017). Additionally, we test confidence correctors with confidence target (CCCT) – we use an MLP with the same architecture as for our method, but instead of using targets as defined in Equation (2), we use:

$$\otimes'_m = \begin{cases} 1, & \text{if } y_m = y \\ 0, & \text{if } y_m \neq y \end{cases}$$

which essentially calibrates the meta-classifier instead of aligning it with EEFP target. The results demonstrate that the improvements provided by our approach are not due to the larger number of parameters of MLP that we use in comparison to calibration methods. While the CCCT variant is effective in calibrating the exits, the overall performance of the model is inferior to that of the one where EEFP is optimized instead.

*Table 9.* Accuracy, ECE, and EEFP score for calibration techniques, our confidence correction, and CCCT correction for ResNet34 evaluated on TinyImagenet.

| | FLOPs threshold | | | ECE($\Downarrow$) | EEFP($\Uparrow$) |
|---|---|---|---|---|---|
| | 25% | 50% | 75% | | |
| Baseline | 48.20 | 62.44 | 65.66 | 0.13 | 0.73 |
| Temperature Scaling | $48.91 \pm 0.01$ | $62.98 \pm 0.01$ | $65.73 \pm 0.00$ | $0.02 \pm 0.00$ | $0.74 \pm 0.00$ |
| Vector Scaling | $48.94 \pm 0.03$ | $62.81 \pm 0.02$ | $66.04 \pm 0.01$ | $0.02 \pm 0.00$ | $0.74 \pm 0.00$ |
| Matrix Scaling | $37.06 \pm 0.19$ | $54.79 \pm 0.05$ | $62.31 \pm 0.15$ | $0.19 \pm 0.00$ | $0.70 \pm 0.00$ |
| Ours top-1 | $48.81 \pm 0.13$ | $62.82 \pm 0.06$ | $66.45 \pm 0.03$ | $0.26 \pm 0.00$ | $0.74 \pm 0.00$ |
| Ours top-2 | $49.86 \pm 0.10$ | $63.79 \pm 0.05$ | $66.94 \pm 0.01$ | $0.26 \pm 0.00$ | $\mathbf{0.78 \pm 0.00}$ |
| Ours top-5 | $\mathbf{50.67 \pm 0.01}$ | $\mathbf{64.06 \pm 0.06}$ | $\mathbf{67.03 \pm 0.03}$ | $0.26 \pm 0.00$ | $\mathbf{0.78 \pm 0.00}$ |
| Ours top-10 | $50.67 \pm 0.08$ | $63.91 \pm 0.08$ | $66.86 \pm 0.02$ | $0.26 \pm 0.00$ | $\mathbf{0.78 \pm 0.00}$ |
| Ours top-200 | $50.57 \pm 0.07$ | $63.36 \pm 0.03$ | $66.39 \pm 0.06$ | $0.26 \pm 0.00$ | $0.76 \pm 0.00$ |
| CCCT top-1 | $48.79 \pm 0.01$ | $62.82 \pm 0.03$ | $66.36 \pm 0.02$ | $0.02 \pm 0.00$ | $0.74 \pm 0.00$ |
| CCCT top-2 | $49.12 \pm 0.05$ | $63.45 \pm 0.05$ | $66.52 \pm 0.02$ | $0.02 \pm 0.00$ | $0.76 \pm 0.00$ |
| CCCT top-5 | $49.12 \pm 0.04$ | $63.49 \pm 0.06$ | $66.58 \pm 0.02$ | $\mathbf{0.01 \pm 0.00}$ | $0.76 \pm 0.00$ |
| CCCT top-10 | $49.18 \pm 0.05$ | $63.46 \pm 0.09$ | $66.56 \pm 0.01$ | $0.02 \pm 0.00$ | $0.76 \pm 0.00$ |
| CCCT top-200 | $49.27 \pm 0.02$ | $63.25 \pm 0.08$ | $66.35 \pm 0.04$ | $0.05 \pm 0.00$ | $0.75 \pm 0.00$ |

# F. Robustness Analysis

To assess the robustness of our approach, we evaluate the models on DomainNet (Peng et al., 2019), which contains the same set of classes across six different domains. We train the model on the real domain and evaluate it across each domain to assess in-distribution versus out-of-distribution performance. The results are presented in Table 10. Our approach is robust to distributional shifts and achieves better scores than calibration in 10 out of 15 cases. We also observe that: (1) temperature scaling actually decreases ECE on unseen domains, and (2) EEFP still correlates well with the overall cost-accuracy trade-off.

*Table 10.* Accuracy, ECE, and EEFP scores on the DomainNet dataset of ResNet-34 EENN, evaluated on the in-domain (Real) and out-of-domain splits.

| | FLOPs threshold | | | ECE($\Downarrow$) | EEFP($\Uparrow$) |
|---|---|---|---|---|---|
| | 25% | 50% | 75% | | |
| **Domain Real** | | | | | |
| Baseline | 58.23 | 76.25 | 78.91 | 0.21 | 0.81 |
| Calibration | $59.01 \pm 0.00$ | $76.91 \pm 0.00$ | $78.90 \pm 0.00$ | $\mathbf{0.02 \pm 0.00}$ | $0.84 \pm 0.00$ |
| Ours | $\mathbf{59.54 \pm 0.02}$ | $\mathbf{77.48 \pm 0.04}$ | $\mathbf{78.96 \pm 0.00}$ | $0.16 \pm 0.00$ | $\mathbf{0.85 \pm 0.00}$ |
| **Domain Clipart** | | | | | |
| Baseline | 18.23 | 28.90 | 36.01 | 0.04 | 0.61 |
| Calibration | $17.86 \pm 0.00$ | $29.35 \pm 0.00$ | $36.35 \pm 0.00$ | $\mathbf{0.18 \pm 0.00}$ | $0.62 \pm 0.00$ |
| Ours | $\mathbf{18.97 \pm 0.01}$ | $\mathbf{30.27 \pm 0.03}$ | $\mathbf{36.47 \pm 0.06}$ | $0.51 \pm 0.00$ | $\mathbf{0.66 \pm 0.00}$ |
| **Domain Infograph** | | | | | |
| Baseline | 5.73 | 10.72 | 14.43 | $\mathbf{0.11}$ | 0.60 |
| Calibration | $\mathbf{5.76 \pm 0.00}$ | $\mathbf{11.00 \pm 0.00}$ | $\mathbf{14.47 \pm 0.00}$ | $0.31 \pm 0.00$ | $\mathbf{0.61 \pm 0.00}$ |
| Ours | $5.61 \pm 0.01$ | $10.51 \pm 0.06$ | $14.26 \pm 0.03$ | $0.68 \pm 0.00$ | $0.60 \pm 0.00$ |
| **Domain Painting** | | | | | |
| Baseline | 17.57 | 26.80 | 33.73 | $\mathbf{0.04}$ | 0.62 |
| Calibration | $17.30 \pm 0.00$ | $26.65 \pm 0.00$ | $\mathbf{33.89 \pm 0.00}$ | $0.20 \pm 0.00$ | $0.64 \pm 0.00$ |
| Ours | $\mathbf{18.16 \pm 0.04}$ | $\mathbf{27.61 \pm 0.05}$ | $33.73 \pm 0.05$ | $0.53 \pm 0.00$ | $\mathbf{0.66 \pm 0.00}$ |
| **Domain Quickdraw** | | | | | |
| Baseline | 1.37 | 1.84 | 2.35 | $\mathbf{0.16}$ | $\mathbf{0.55}$ |
| Calibration | $1.39 \pm 0.00$ | $1.88 \pm 0.00$ | $\mathbf{2.42 \pm 0.00}$ | $0.34 \pm 0.00$ | $\mathbf{0.55 \pm 0.00}$ |
| Ours | $\mathbf{1.48 \pm 0.00}$ | $\mathbf{2.00 \pm 0.02}$ | $2.38 \pm 0.02$ | $0.76 \pm 0.00$ | $0.54 \pm 0.00$ |
| **Domain Sketch** | | | | | |
| Baseline | 9.27 | 17.33 | 22.50 | $\mathbf{0.04}$ | 0.58 |
| Calibration | $9.32 \pm 0.00$ | $17.54 \pm 0.00$ | $22.68 \pm 0.00$ | $0.23 \pm 0.00$ | $0.59 \pm 0.00$ |
| Ours | $\mathbf{9.58 \pm 0.01}$ | $\mathbf{18.08 \pm 0.08}$ | $\mathbf{23.09 \pm 0.01}$ | $0.61 \pm 0.00$ | $\mathbf{0.62 \pm 0.00}$ |

# G. Generalization to Other Early-Exit Methods

In this section, we extend our evaluation to include other early exit systems. Our failure prediction-based method can be applied to any early-exit method that uses classifier confidence for exit decisions. Since our approach is complementary to these methods, we apply our method to each of these models to explore whether the performance improvements that we observed in the main paper do not diminish. In Table 11 we report the results of our method applied to Gradient Equilibrium (Li et al., 2019), Zero Time Waste (Wójcik et al., 2023), and early-exit distillation (Phuong & Lampert, 2019). Our approach provides consistent gains in the overall performance of the network in all cases. Crucially, our method also consistently achieves better EEFP scores while being miscalibrated compared to temperature scaling. This confirms our hypothesis and strengthens the main results of our work.

*Table 11.* Accuracy, ECE, and EEFP scores for three ResNet34-based EENN models, evaluated on the TinyImagenet dataset.

| | FLOPs threshold | | | ECE($\Downarrow$) | EEFP($\Uparrow$) |
|---|---|---|---|---|---|
| | 25% | 50% | 75% | | |
| | Gradient Equilibrium | | | | |
| Baseline | 46.07 | 59.59 | 64.01 | 0.12 | 0.72 |
| Calibrated | $46.39 \pm 0.00$ | $60.33 \pm 0.01$ | $64.28 \pm 0.01$ | $\mathbf{0.02 \pm 0.00}$ | $0.73 \pm 0.00$ |
| Ours | $\mathbf{47.70 \pm 0.05}$ | $\mathbf{61.31 \pm 0.03}$ | $\mathbf{64.87 \pm 0.05}$ | $0.07 \pm 0.00$ | $\mathbf{0.77 \pm 0.00}$ |
| | Zero Time Waste | | | | |
| Baseline | 45.80 | 59.04 | 62.22 | 0.13 | 0.71 |
| Calibrated | $46.34 \pm 0.01$ | $59.45 \pm 0.01$ | $62.38 \pm 0.01$ | $\mathbf{0.01 \pm 0.00}$ | $0.72 \pm 0.00$ |
| Ours | $\mathbf{48.00 \pm 0.05}$ | $\mathbf{60.03 \pm 0.04}$ | $\mathbf{63.24 \pm 0.03}$ | $0.08 \pm 0.00$ | $\mathbf{0.75 \pm 0.00}$ |
| | Early-Exit Distillation | | | | |
| Baseline | 47.35 | 61.15 | 63.84 | 0.13 | 0.73 |
| Calibrated | $48.11 \pm 0.00$ | $61.70 \pm 0.00$ | $64.21 \pm 0.00$ | $\mathbf{0.02 \pm 0.00}$ | $0.74 \pm 0.00$ |
| Ours | $\mathbf{49.33 \pm 0.02}$ | $\mathbf{62.47 \pm 0.04}$ | $\mathbf{65.00 \pm 0.01}$ | $0.07 \pm 0.00$ | $\mathbf{0.78 \pm 0.00}$ |

# H. Contributions

**Piotr Kubaty** led the empirical investigation, conducting the majority of the experiments and generating the data visualizations. He conceptualized the architectural refinements for the meta-classifier, specifically the top-k selection mechanism. He also made significant contributions to the manuscript.

**Filip Szatkowski** served as the primary coordinator for the writing process. He managed the structural development of the manuscript across multiple versions and was responsible for ensuring narrative clarity and logical flow throughout the paper.

**Grzegorz Choczyński** conducted the experimental evaluations demonstrating the generalization of the proposed approach to other early-exit frameworks (Table 11). He also performed several additional experiments that were not included in the final manuscript.

**Eric Nalisnick** developed the theoretical framework of the paper. He authored the formal analysis regarding the interplay between calibration, EENN triviality, and the conditional anytime property (Section 2.2 and Section 3) and provided crucial refinements to the final manuscript.

**Bartosz Wójcik** conceptualized the core research direction and supervised the project. He identified the fundamental disconnect between local exit calibration and global EENN performance, explicitly drawing the connection to the failure prediction literature. He defined the optimal stopping decision for EENNs, formulated the Early-Exit Failure Prediction (EEFP) score, and proposed the meta-classifier approach that directly optimizes for this target.

