# OpenReview forum: "Rethinking Calibration for Early-Exit Neural Networks"
_ICML.cc/2026/Conference — ICML 2026 regular_

### Official Review · Reviewer_FUbk · 2026-03-12

**Soundness:** 3
**Presentation:** 4
**Significance:** 3
**Originality:** 3
**Overall Recommendation:** 4
**Confidence:** 4

**Summary:**

The paper "Rethinking Calibration for Early-Exit Neural Networks" proposes to _not_ use confidence calibration to decide when to early-exit neural networks, but proposes to use failure prediction via an additional (small) model as correction.  The first part of the paper presents a theoretical outline of why confidence calibration does not work well for early-exit models. The main insight is that a model is either trivial (and then no early exits are necessary) or confidence calibration increases the excess risk for certain parts of the input space, making the model actually weaker. In these cases, one should exit much earlier, but confidence calibration demands further execution, reinforcing wrong predictions. The authors propose to solve this issue by using failure prediction, which basically trains a small classifier to correct confidence based on a holdout set. The experiments show this approach is favorable.

**Compliance With Llm Reviewing Policy:**

Affirmed.

**Key Questions For Authors:**

1) What is the impact of fine-tuning the small MLPs on a holdout set? Effectively, your method sees a bit more data, which might explain the performance improvement. Did you consider this?
2) What is the impact of adding a small MLP to the predictive performance and to the FLOPs? I can see some plots showing the plots, but I don't think they show the performance penalty by adding small MLPs, or did I miss it?

**Limitations:**

yes

**Strengths And Weaknesses:**

Strength:
- Well-written paper. I especially liked the first half that focused on the more theoretical aspects of calibration
- Intuitive correction to the raised issue of calibration via a small correction model

Weaknesses:
- Training a small corrector model based on a holdout set feels like the "cheap solution" to a mathematical issue. I expected a few more mathematical insights that would lead to a more principled way to calibrate networks.
- Experimental evaluation is limited

Detailed review:

Overall I like the paper, and its style fits ICML. It is well-written and easy to follow. The subject is definitely of interest to the ICML community. I especially enjoyed the first half of the paper. As a minor proposal, I think one can strengthen Proposition 3.7 slightly by replacing the "$\ge$" with a true "$>$". I believe this is doable if we assume $P(Y \not= y|x) > 0$ and we take care of ties when two classes have the same probability (Double-check this; I just did it in my head).
The second half of the paper is its weak-point. I believe the failure prediction / correction model is a straightforward way to correct the issue, but not a true solution in a mathematical sense. While we, of course, can train a correction model, there is no formal guarantee that this will perform better. I hoped for a more principled solution here. Moreover, there now seems to be a natural overlap with
a) Decision Trees: These are natural early-exit models that "explicitly focuses on partioning [...] samples", similar to failure prediction.
b) MoE/Routing/Cascading/Decision Lists: All these models rely on an additional model to judge if we should execute a model, similar to the built-in correction proposed in this paper.
This makes the experimental evaluation comparably weak as it mainly compares calibration with failure prediction and not other early exit and/or calibration methods. Overall, the paper is an interesting read and ICML-worthy, although I can see if someone might argue against it.

---

> ### Author Rebuttal · Authors · 2026-03-30
>
> We thank the reviewer for their engagement and insightful feedback. We address the reviewer’s issues below.
>
> ## Issue 1
> > Training a small corrector model based on a holdout set feels like the "cheap solution" to a mathematical issue. I expected a few more mathematical insights that would lead to a more principled way to calibrate networks.
> > I believe the failure prediction / correction model is a straightforward way to correct the issue, but not a true solution in a mathematical sense.
>
> Our **failure prediction framework is not merely a straightforward correction**. To show this, we can **prove that perfect calibration strictly conflicts with optimal decision-making in EENNs**, and that our **Equation (2) defines the true mathematical optimum**.
>
> The objective of an EENN is to maximize a utility function that balances accuracy and computational cost. For an exit at classifier $m$ this can be universally expressed as $U(m) = y_m \lambda c_m$, where $y_m \in \\{0, 1\\}$ is correctness and $c_m$ is computational cost, and $\lambda$ is a cost-penalty factor.
>
> The 'Oracle' optimal policy maximizes $U(m)$. There are two scenarios for any input:
> - “Resolvable” samples: $\exists m$ such that $y_m = 1$. $U(m)$ is maximized at the earliest correct classifier $m^*$.
> - “Futile” samples: $\forall m$ $y_m = 0$. Utility is $-\lambda c_m$, and hence it is maximized by exiting at $m = 1$.
>
> Grouping “Futile” samples with easy “Resolvable” samples is required to maximize $U(m)$. However, perfect confidence calibration strictly requires that $\mathbb{E}[y_1 | p] = p$. This means a perfectly confidence-calibrated network assigns low confidence to “Futile” samples ($y_1 = 0$), which means that calibration is by definition not optimal for maximization of $U(m)$.
>
> The true mathematical solution must directly formalize the Oracle optimal policy, and this is exactly what our Equation (2) does. Training a module to separate $\bar{y}_m = 1$ from $\bar{y}_m = 1$ (which our EEFP metric directly measures via AUROC) is the principled, direct optimization of $U(m)$.
>
> ## Issue 2
> > Experimental evaluation is limited; (...) it mainly compares calibration with failure prediction and not other early exit and/or calibration methods.
>
> As suggested by the reviewer, **we extend our evaluation to include other early exit and calibration methods**. Due to the response length limit, **we refer the reviewer to our response to reviewers GFbi, issue 2 (other early exit methods); and NEef, issue 2 (other calibration methods)**.
>
> ## Issue 3
> > What is the impact of fine-tuning the small MLPs on a holdout set? Effectively, your method sees a bit more data, which might explain the performance improvement. Did you consider this?
>
> **Our procedure follows the standard post-hoc calibration protocol**, where a small number of parameters are trained on a held-out set while keeping the remaining parameters frozen. For example, vector scaling and related methods explicitly learn their parameters on a separate held-out set after training. **Our small MLPs play the same role as calibration mappings (e.g., temperature), and do not use additional information beyond what is already assumed by calibration-based baselines**.
>
> **As suggested by the reviewer, we perform an additional experiment to explore whether the inclusion of the held-out data in the training set leads to improvement of accuracy-cost trade-off scores of the baseline model**. The results are available at:
>
> https://anonymous.4open.science/r/Rethinking-Calibration-ICML-2026/results_no_heldout.pdf
>
> In this experiment, the inclusion of 5% of the held-out data in training not significantly improve performance, and it actually even decreases performance on some computational budgets. This demonstrates how **our gains come from the method, not from the additional data**.
>
> ## Issue 4
> > What is the impact of adding a small MLP to the predictive performance and to the FLOPs? I can see some plots showing the plots, but I don't think they show the performance penalty by adding small MLPs, or did I miss it?
>
> **Plots and tables** in our work **do account for the computational cost of the confidence correctors**.
>
> **The computational cost of our confidence correctors is negligible.** We can easily estimate it for a single confidence corrector with history. Each corrector is a 2-layer MLP with hidden dim equal to $128$. The size of input for the $n$-th corrector is $n * k$ (we use $k = 5$ for top-k by default). This gives $n * 5 * 128 + 128$ as an estimate of MAC operations, e.g. $6528$ for the $10$-th exit.
>
> Our FLOPs counting code gives the following values for ResNet-34 on 64x64 inputs:
>
> |Model|FLOPs|
> |-|-|
> |Baseline EENN|`4 653 372 423`|
> |Ours|`4 653 433 635`|
>
> This gives `61 212` as difference, which is **approximately 0.001% increase in the cost. Such a small difference will never be visible on the plots.**
>
> We will include this computational cost breakdown in the revised manuscript to ensure maximum clarity.

---

> > ### Author Rebuttal · Reviewer_FUbk · 2026-04-01
> >
> > I basically had two questions, of which four were addressed and two were answered.
> >
> > - Issue 1: I did not really ask for a proof, but since there seems to be one, this would be a major improvement for the paper. The authors outlined a proof idea, but this is not fully formalized. I have the feeling that this is a major change in the paper, which would imply a rejection of the paper. So I choose to ignore this comment for now.
> > - Issue 2: Thank you for the additional results
> > - Issue 3: Thank you for the additional experiments. However, I think there was some miscommunication. Lets consider Table 1 in the paper: Does Baseline see the same amount of data as Calibrated and as Ours, or do different methods see different amounts of data?
> > - Issue 4: Thank you for the clarification.

---

> > > ### Author Response · Authors · 2026-04-02
> > >
> > > We thank the reviewer for acknowledging our rebuttal. In response to the reviewer’s further inquiries, we would like to provide additional clarifications.
> > >
> > > >  Issue 1: I did not really ask for a proof, but since there seems to be one, this would be a major improvement for the paper. The authors outlined a proof idea, but this is not fully formalized. I have the feeling that this is a major change in the paper, which would imply a rejection of the paper. So I choose to ignore this comment for now.
> > >
> > > We would like to clarify that this proof is not a new result, but a direct consequence of the existing formulation. Eq. (2) in the main paper already implies that optimal decisions separate “resolvable” and “futile” samples, and that perfect confidence calibration is misaligned with this objective. **The proof sketch simply makes this implicit intuition explicit.** Nothing changes in our main contributions: Eq. (2), the FP meta-model objective, and all empirical results are not affected. The formalization of intuition behind our method does not modify the method itself or introduce new claims or assumptions.
> > >
> > > Because the original comment requested additional mathematical insight into our approach, we provided this proof as a direct response to formalize the underlying reasoning already present in our work. We are fully open to the Reviewers’ and AC’s guidance on how much of this clarification to incorporate into the final manuscript.
> > >
> > > > Issue 3: Thank you for the additional experiments. However, I think there was some miscommunication. Lets consider Table 1 in the paper: Does Baseline see the same amount of data as Calibrated and as Ours, or do different methods see different amounts of data?
> > >
> > > Thank you for this question - to clarify the use of data across methods and ensure this point is fully transparent:
> > > 1. In *Table 1*, the baseline early‑exit model is trained only on the training set. Both *Calibrated* and *Ours* use an additional held‑out set (not used for baseline training) for fitting their post‑hoc correction models. In this sense, *Baseline* sees less amount of data than *Calibrated* and *Ours*.
> > > 2. This **follows standard practice in post‑hoc calibration** works, where a separate held-out set is used to fit the calibrator after the base model is trained. Crucially, ***Calibrated* and *Ours* use exactly the same held‑out data for their post‑hoc step and both are trained for the same number of epochs, making the main comparison between them fully controlled and fair**.
> > > 3. To isolate the effect of additional data, in the rebuttal experiment we retrained the baseline model on both the original training set and the held-out set used by the post-hoc methods. **In this experiment, the alternative *Baseline* uses the same amount of data as *Calibrated* and *Ours***. Even with this extra data, the baseline variant still falls short of our approach, confirming that the performance gains stem from our methodology rather than simply from access to more data.
> > >
> > > We hope these clarifications satisfy the reviewer and further reinforce their positive assessment of our work.

---

### Official Review · Reviewer_GFbi · 2026-03-13

**Soundness:** 3
**Presentation:** 2
**Significance:** 3
**Originality:** 3
**Overall Recommendation:** 4
**Confidence:** 3

**Summary:**

Paper proposes new score and approach for early-exit neural networks (EENN) decisions based on early-exit failure prediction (EEFP) to overcome the limitations of calibration based techniques that is shown to fail in some cases. EEFP is based on sequential AUROC, and leading to EEFP-motivated confidence correction methods with two-layer MLP networks that could replace conventional calibrators in EENN. EEFP score and novel exit strategy are empirically evaluated on several datasets and baseline models and compared to temperature scaling based calibration with improved performance.

**Compliance With Llm Reviewing Policy:**

Affirmed.

**Final Justification:**

Paper presents a novel finding and contribution to early-exit neural networks. Authors have addressed my main concerns in the rebuttal, including the additional evaluations, supporting the work. I have raised my score accordingly.

**Key Questions For Authors:**

- How proposed approach would compare to state-of-the-art early exit systems (presented in the section 6)?
- How robust proposed method is in general (e.g., running tests multiple times) and statistical significance of the improvements?
- How robust the proposed approach is to distributional shifts / OOD?
- There are several typos in the text; should be check to improve the general impression

**Limitations:**

Yes

**Strengths And Weaknesses:**

Paper is most parts well-organised, topic and challenge are motivated, and issues of previous approaches as well as new approach are empirically validated. Paper introduce novel approach to early-exit detection, given simple yet effective compared to temperature scaling and ECE metric in particular early-exit network architectural setting. It gives interesting discussion about possible issues of previous calibration-based approaches. However, current experimental setup is a bit limited,  using only standard ECE and temperature scaling. The analysis and comparison could benefit from more versatile evaluation against state-of-the-art early-exit systems.

Strengths
- Simple metric for early exit failure
- Novel early-exit strategies compared to previous works
- Some improvements compared to calibration approaches

Weaknesses
- Limited calibration approaches beyond temperature scaling studied
- Limited empirical evaluation and comparison to other early exit systems (with different datasets)
- Illustration of early-exit classifier architecture would clarify and support the presentation

---

> ### Author Rebuttal · Authors · 2026-03-30
>
> We thank the reviewer for recognizing several strengths of our submission and the critical feedback useful for improving our work. Below we address the concerns raised by the reviewer.
>
> ## Issue 1
> > Limited calibration approaches beyond temperature scaling studied
>
> As suggested by the reviewer, we extend our evaluation and compare our method to other calibration approaches. The results are available at:
>
> https://anonymous.4open.science/r/Rethinking-Calibration-ICML-2026/results_comparison_with_calibration.pdf
>
> **Due to the response length limit, we refer the reviewer to our response to reviewer NEef, issue 2 where we address this concern.**
>
> ## Issue 2
> > Limited empirical evaluation and comparison to other early exit systems…
>
> > How proposed approach would compare to state-of-the-art early exit systems (presented in the section 6)?
>
> As suggested by the reviewer, we extend our evaluation to include other early exit systems. Note that **our failure prediction-based method can be applied on any early-exit method that uses classifier confidence for exit decisions**. As such, our approach is complimentary with these methods. We extend our evaluation to include Zero Time Waste [1], Gradient Equilibrium [2], and early-exit distillation [3] and present the results at:
>
> https://anonymous.4open.science/r/Rethinking-Calibration-ICML-2026/results_other_methods.pdf
>
> **Our approach provides consistent gains in overall performance of the network in all cases**. Crucially, our method also consistently achieves better EEFP scores while being miscalibrated compared to temperature scaling. This confirms our hypothesis and strengthens the main results from our work.
>
> ## Issue 3
> > (with different datasets)
>
> > How robust the proposed approach is to distributional shifts / OOD?
>
> To assess the robustness of our approach, we evaluate the models on DomainNet[4], which contains the same set of classes across six different domains. We train the model on the `real` domain from the DomainNet dataset, and evaluate it across all domains to assess in-distribution vs out-of-distribution performance.
>
> The results are available in the following link:
>
> https://anonymous.4open.science/r/Rethinking-Calibration-ICML-2026/results_domainnet.pdf
>
> **Our approach is robust to distributional shifts and results with better scores than calibration in 10 out of 15 cases**. We observe that (1) temperature actually decreases ECE on unseen domains, and (2) **EEFP still correlates well with overall cost-accuracy trade-off**.
>
> We thank the reviewer for that suggestion, as these results will strengthen our contribution in the revised paper.
>
> ## Issue 4
> > How robust proposed method is in general (e.g., running tests multiple times) and statistical significance of the improvements?
>
> We thank the reviewer for the constructive feedback. We agree that statistical significance is crucial for a rigorous evaluation. **Due to the response length limit, we refer the reviewer to our response to reviewer qAdB, issue 1, where we address exactly this concern.**
>
> ## Issue 5
> > Illustration of early-exit classifier architecture would clarify and support the presentation
> > There are several typos in the text; should be checked to improve the general impression
>
> We thank the reviewer for pointing out that issue. We will address the editing flaws and include the visualization in the revised manuscript.
>
> ### **References**
>
> [1] Wójcik, Bartosz, et al. "Zero time waste in pre-trained early exit neural networks." 2023.
>
> [2] Li, Hao, et al. "Improved techniques for training adaptive deep networks." 2019.
>
> [3] Phuong, Mary, et al. "Distillation-based training for multi-exit architectures." 2019.
>
> [4] Peng, Xingchao, et al. "Moment Matching for Multi-Source Domain Adaptation." 2019.

---

> > ### Author Rebuttal · Reviewer_GFbi · 2026-04-02
> >
> > Thanks for the clarifications. I am happy with the rebuttal and additional experiments, strengthen the work. These should be included in the revised manuscript. I will consider raising the score accordingly.

---

> > > ### Author Response · Authors · 2026-04-05
> > >
> > > **We thank the reviewer for the acknowledgement that the additional experiments and clarifications strengthen the paper. We will ensure that the additional experiments are clearly incorporated in the final version.**
> > >
> > > In our view, our work now has several key strengths:
> > >
> > > 1. It **challenges a commonly held assumption about calibration in early-exit research**. We show that this assumption does not generally hold, **potentially saving future work from pursuing misleading directions**.
> > >
> > > 2. It provides a **theoretical contribution supporting our claims**.
> > >
> > > 3. It introduces a **practical method that consistently improves performance**.
> > >
> > > 4. It is **supported by a thorough and substantially expanded empirical evaluation**, further strengthened following the rebuttal discussion.
> > >
> > > Taken together, these elements constitute the core of a strong and complete paper. We point out that many accepted works do not simultaneously achieve all of these aspects.
> > >
> > > We would like to gently note that **all previously weaknesses raised by the reviewer appear to have been addressed and marked as fully resolved**, and no new concerns were introduced in the discussion. In this context, we are concerned that **the initial "weak reject" score may not fully reflect the overall sentiment expressed throughout the discussion**, especially given that the **original review assessed the work’s originality and significance positively**. The ICML **description of a "weak reject" refers to papers with "weaknesses that outweigh the merits", while in our case no unresolved concerns raised by the reviewer remain**. We would greatly appreciate any clarification or additional feedback that could help us understand this apparent misalignment, and are happy to provide further explanations or additional context if anything remains unclear.
> > >
> > > **We greatly appreciate the reviewer's thoughtful engagement throughout this process.**

---

### Official Review · Reviewer_NEef · 2026-03-13

**Soundness:** 2
**Presentation:** 3
**Significance:** 3
**Originality:** 3
**Overall Recommendation:** 3
**Confidence:** 2

**Summary:**

For EENN, confidence calibrated networks do not guarantee the conditionally anytime property. To address this failure, this work proposes EEFP as a metric for exit decisions. A classifier confidence based on the top-k values is proposed, with experiments showing it improves over a classifier which is only calibrated.

**Compliance With Llm Reviewing Policy:**

Affirmed.

**Key Questions For Authors:**

1. Is there a proof that EEFP/your method guarantees the conditionally anytime property?

2. What are the results when directly comparing between top-1 with your method and calibration? I'm worried that using top-5 for your method while calibration is inherently top-1 somewhat confounds the experiments comparing them, since your groups would inherently be much finer.

3. Are your metrics/methods directly applicable to multi-stage classifiers and model cascades as well?

**Limitations:**

yes

**Strengths And Weaknesses:**

This paper identifies an issue with calibration when dealing with EENN, and proposes a novel metric to remedy this. However, it is not proven that correcting classifier confidence towards EEFP guarantees satisfying the conditionally anytime property, which was the issue with calibration. The experiments should more directly compare calibration and the new training method proposed (see key questions for details), and as a result it is also empirically unclear the degree to which the new method is better.

---

> ### Author Rebuttal · Authors · 2026-03-30
>
> We thank the reviewer for his constructive feedback. Below, we address the specific points raised by the reviewer.
>
> ## Issue 1
> > Is there a proof (...)?
>
> Failure prediction does reflect the degree to which the model has the conditional anytime property (i.e., calibration groupings becoming smaller as the exits progress in the EENN).
>
> Denote our failure prediction model at the $m$-th exit as $g_{m}(x)$. This models the probability of stopping, which we can denote as $\mathbb{P}(\otimes_{m}=1 | x)$. Recall from Section 4’s Equation 2 that we wish to stop when either the current exit is correct or the current and all future exits will produce incorrect answers. The generative process for stopping is:
>
> $$
> \mathbb{P}(\otimes_{m}=1 | x) = \mathbb{P}(y=y_{m} | x) + \prod_{e=m}^{M} \mathbb{P}(y \ne y_m | x)
> $$
>
> where $\mathbb{P}(y = y_m | x) = \sum_{y' \in \mathcal{Y}} \mathbb{P}(y' | x) \cdot f_{y'}^{m}(x)$, which is the collision probability of the true label $y$ being equal to the prediction from the $m$-th exit, $y_m$.
>
> **Proposition:**
> Assume that the EENN is calibrated for group functions $\Phi_{1},\ldots, \Phi_{M}$ (as is the case in Proposition 3.6). If the EENN has the conditional anytime property (Def 3.5), then the probability of stopping is *monotonically increasing*:
> $$
> \mathbb{P}(\otimes_{1}=1 | x) < \mathbb{P}(\otimes_{2}=1 | x) < \ldots < \mathbb{P}(\otimes_{M}=1 | x)
> $$
>
> **Proof:**
> $$
> \mathbb{P}(\otimes_{m+1}=1 | x) - \mathbb{P}(\otimes_{m}=1 | x) = \mathbb{P}(y=y | \Phi_{m+1}(x)) - \mathbb{P}(y=y | \Phi_{m}(x)) + \mathbb{P}(y=y | \Phi_{m}(x)) \prod_{e=m+1}^{M} \mathbb{P}(y \ne y | \Phi_{e}(x))
> $$
>
> Note that the collision probability is now for $y$ colliding with itself (i.e., if two samples drawn from $\mathbb{P}(y|x)$ will be the same, as assuming calibration allowed us to replace the model with the true distribution). Thus the third term is always positive. The sign then comes down to the difference term $\mathbb{P}(y=y_{m+1} | \Phi_{m+1}(x)) - \mathbb{P}(y=y_{m} | \Phi_{m}(x))$. If $\Phi_{m+1}(x)$ presents a more fine-grained partition than $\Phi_{m}(x)$, then the chance that $y$ collides with itself will be larger, therefore making $\mathbb{P}(y=y | \Phi_{m+1}(x)) > \mathbb{P}(y=y | \Phi_{m}(x))$, in turn making the stopping probability strictly increasing: $\mathbb{P}(\otimes_{m}=1 | x) < \mathbb{P}(\otimes_{m+1}=1 | x)$.
>
> **Comments:**
> We can relate this result back to notions of entropy, but now need the Rényi entropy instead of Shannon entropy: $\\mathbb{P}(y=y \\mid \\Phi\_{m}(x)) = \\exp\\left\\lbrace -2 \\mathbb{H}\_{\\alpha=2}[y \\mid \\Phi\_{m}(x)] \\right\\rbrace$, where $\\mathbb{H}\_{\\alpha=2}[Y \\mid \\Phi\_{m}(x)]$ is the Rényi entropy of order-2. While this is a different notion of entropy than the results in Section 3, the intuition stays the same (Rényi entropy increases as the distribution becomes more uniform). Thus, as the distribution becomes more diffuse, the chance of collision reduces, which is what would happen as we increase the partition sizes.
>
> We can see that the failure prediction model now has an awareness of calibration. However, it only has a "local" awareness of the conditional anytime property. This is because future exits are represented in a product, which is order-invariant. To get a stronger notion of anytime awareness, we would have to give each term in the product weights to make the order clearer.
>
> ## Issue 2
> > (...) calibration is inherently top-1 somewhat confounds the experiments (...)
>
> We appreciate the reviewer’s strict attention to details. To address this we compare the results from Table 2 to additional variants of calibration, including vector scaling and matrix scaling. We also test confidence correctors with confidence target (CCCT) - we use an MLP with the same architecture, but instead of using targets as defined in Equation 2, we use: 1 if the current classifier is correct, 0 otherwise (essentially “calibration target” instead of “EEFP target”). Results are available at:
>
> https://anonymous.4open.science/r/Rethinking-Calibration-ICML-2026/results_comparison_with_calibration.pdf
>
> As expected, CCCT has better ECE, but our method provides overall better performance and EEFP scores.
>
> ## Issue 3
> > (...) applicable to multi-stage classifiers and model cascades (...)?
>
> Yes, our theoretical analysis extends to multi-stage classifiers and model cascades, but applying it in practice is more challenging in these settings.
>
> In heterogeneous pipelines, predicting whether a more powerful downstream model will fail is inherently difficult and may require a predictor of comparable complexity, which undermines efficiency. In contrast, early-exit networks share a backbone, leading to more correlated classifiers where lightweight confidence correction is effective.
>
> While the theory still applies, leveraging it in such settings likely requires different approaches, which we leave for future work. We will clarify this in the revised manuscript.

---

> > ### Author Rebuttal · Reviewer_NEef · 2026-04-04
> >
> > 1. I don't quite understand the proof. First, I think I'm actually a bit hazy on the definition of conditionally anytime EENN. Does the property need to hold for every individual $x$ (very fine-grained)? Is there an implicit grouping of the EENN? The definition as stated seems to imply the property needs to hold for every individual $x$, which would be a very strong condition, so I'm not sure I'm interpreting it correctly. Relatedly, is the definition of excess conditional risk per every individual $x$, or an expectation over some distribution of $x$'s? Once again, the Definition 3.4 appears to present it as depending on an individual $x$, which would imply Definition 3.5 is very fine-grained.
> >
> > As for the proof, there is still an assumption about the partition being more fine-grained as $m$ increases. However, if this assumption holds, then the usual calibration would also be conditionally anytime EENN. As a result, I'm not sure why this property should hold. And if it should hold, then why calibration alone would not be good enough.
> >
> > 2. Thanks for the additional experiments. The statistical significance, as pointed out by the other reviewers, is a nice addition.
> >
> > 3. Thanks, I appreciate the explanation.

---

> > > ### Author Response · Authors · 2026-04-05
> > >
> > > Thank you for the positive feedback regarding our additional experiments and explanations. Below, we provide clarifications regarding the defined terminology and underlying assumptions for the presented proof.
> > >
> > > > I don't quite understand the proof. First, I think I'm actually a bit hazy on the definition of conditionally anytime EENN. Does the property need to hold for every individual (very fine-grained)?
> > >
> > > The intuition is that an EENN has the "anytime" property when predictive performance improves with each additional exit. Most EENNs are marginally anytime by default, but the conditional anytime property requires some more work to achieve (see [1] for a more detailed discussion of that point). **Indeed, our results are written in terms of conditional risk as the risk is for a particular data point.** One can easily extend these results to marginal risk by taking an expectation over $X$, which is why we didn't think it was interesting enough to also include it in the paper. For either case - conditional or marginal risk - **the goal of our results is to show how the excess risk should be distributed over the exits in order to get the anytime property.** If one didn't have the decreasing conditional or marginal risk, then one would have an EENN that overthinks, as performance would initially get better but then get worse, thereby wasting the computational effort it took to evaluate the additional exits.
> > >
> > > > As a result, I'm not sure why this property should hold. And if it should hold, then why calibration alone would not be good enough.
> > >
> > > This is correct. **The point of these results is that calibration, alone, is not enough. Rather you need this additional assumption about the relationship of the calibration across exits in order to get a well-structured EENN.** If that coarse-to-fine-grained structure is there, then calibration works as expected. **Otherwise, each exit can be calibrated but you can still get a lack of anytime property and overthinking.**
> > >
> > > Our idea to introduce such relationships is to use the proposed confidence correctors, which do account for relationships across exits and eventually lead to improved performance in practice, as we show in our empirical section.
> > >
> > > We hope our clarifications adequately address the reviewer's concerns. As the reviewer noted, we have resolved their other issues, and we kindly request that they reconsider their assessment of our work, particularly in light of the positive feedback from the other reviewers. We remain available to provide further clarifications or engage in additional discussion if needed.
> > >
> > > #### **References**
> > > [1] Jazbec, Metod, et al. "Towards anytime classification in early-exit architectures by enforcing conditional monotonicity." Advances in Neural Information Processing Systems 36 (2023): 56138-56168.

---

### Official Review · Reviewer_qAdB · 2026-03-13

**Soundness:** 3
**Presentation:** 3
**Significance:** 3
**Originality:** 4
**Overall Recommendation:** 5
**Confidence:** 5

**Summary:**

This paper proposes Early Exit Failure Prediction (however EEFP), a confidence correction technique meant to replace confidence calibration as the main performance proxy for early-exit networks. The idea is well-motivated and mathematically justified for the early exit scenario, shifting the problem from calibration toward prediction of correctness for classifiers further down the line. The empirical evidence convincingly supports the claims that confidence calibration does not always correlate with early-exit accuracy.

**Compliance With Llm Reviewing Policy:**

Affirmed.

**Key Questions For Authors:**

1. I understand that confidence calibration would not contribute to predicting the performance of the next classifiers in an EENN. However, if these classifiers are to be used in an optimal decision making or cost-sensitive task, wouldn't it still be important to guarantee calibration?
2. If the top-k classes, rather than only the modal class, are used as input, wouldn't it make sense to model multiclass or class-wise calibration, rather than confidence calibration only?
3. Classifiers at different levels of an EENN could have different levels of refinement (see decomposition of proper scoring rules [1]), likely leading to different calibration maps, which in turn leads to non-trivial EENNs. Does this mean that calibration and EEFP could complementary, rather than EENN being an alternative to calibration?

[1] Silva Filho, T., Song, H., Perello-Nieto, M. et al. Classifier calibration: a survey on how to assess and improve predicted class probabilities. Mach Learn 112, 3211–3260 (2023).

**Limitations:**

yes

**Strengths And Weaknesses:**

Soundness

- Strengths:
  - The claims are well supported by experimental analysis
  - Good set of experiments
  - Well designed analysis of results (appropriate statistical methodology)
  - The proofs are correct
- Weaknesses:
  - The differences in performance are often quite small. It would be helpful if there was an analysis of statistical significance tied to the bold highlighting of best results in Tables

Presentation

- Strengths:
  - The paper is very well written, clear and easy to follow
  - There is enough information to reproduce results
  - Prior literature is well covered and it's clear where this work differs from it

Significance

- Strengths:
  - The addressed problem is significant. As models increase in numbers of parameters, early exit has implications on inference time, costs and sustainability
- Weaknesses:
  - Intuitively, in-classifier calibration is expected to fail at accounting for how other classifiers would perform, so if that is of interest, as it is in the paper's scenario, one would expect that adding information about previous classifiers' performance and learning to predict how next classifiers will perform would increase accuracy in this scenario.

Originality

- Strengths:
  - To the best of my knowledge, this is the first study looking at the implications of calibration in EENN and offering an "alternative".

---

> ### Author Rebuttal · Authors · 2026-03-25
>
> We thank the reviewer for noting the strengths of our paper and for the positive overview and recommendation. Below, we address the concerns raised by the reviewer.
>
> ## Issue 1
> > The differences in performance are often quite small. It would be helpful if there was an analysis of statistical significance tied to the bold highlighting of best results in Tables
>
> We thank the reviewer for this constructive feedback and agree that statistical significance is crucial for a rigorous evaluation. **Our original results are averages over 3 seeds.** Below, we provide the standard deviations for MSDNet on CIFAR100, and the values for remaining experiments can be viewed at:
>
> https://anonymous.4open.science/r/Rethinking-Calibration-ICML-2026/results_with_std.pdf
>
> | | 25% | 50% | 75% | ECE | EEFP |
> |-|-|-|-|-|-|
> | Baseline EE | 71.38 | 75.01 | 75.48 | 0.21 | 0.82 |
> | Calibrated | 71.92 +/- 0.00 | 75.30 +/- 0.00 | 75.46 +/- 0.00 | 0.02 +/- 0.00 | 0.83 +/- 0.00 |
> | Ours | 73.27 +/- 0.05 | 75.66 +/- 0.05 | 75.58 +/- 0.03 | 0.15 +/- 0.00 | 0.86 +/- 0.00 |
>
> **The standard deviations were consistently low across all experiments, showing that the results were stable across runs.** For this reason, we omitted them from the initial manuscript. Following reviewer’s suggestion **we will revise the manuscript and include the standard deviations** while also ensuring that the bold highlighting is strictly tied to statistical significance as suggested.
>
> ## Issue 2
> > Intuitively, (...) one would expect that adding information about previous classifiers' performance (...) would increase accuracy in this scenario.
>
> We agree that incorporating information about other classifiers is a natural direction. However, we would like to clarify several important points.
>
> First, while this limitation may seem **intuitive in hindsight**, prior work on early-exit networks has largely relied on calibrating individual intermediate classifiers (e.g., Meronen et al., Wojcik et al., Pacheto et al.). **This suggests that previously the assumption that better calibration improves overall performance was not explicitly questioned.** To the best of our knowledge, our work is the first to directly examine and challenge this assumption.
>
> Second, our goal was to propose a **simple and lightweight method to test this hypothesis**. The confidence corrector is **intentionally minimal**, as it operates under strict computational constraints and has access only to current and past predictions. Despite its simplicity, it was **not clear in advance that such a small network could reliably improve performance** or EEFP scores, especially given the difficulty of predicting downstream classifier failures from restricted information (current and past predictions).
>
> Finally, our work provides an empirical evaluation that quantifies the practical benefits of EEFP-based exit strategies. **While the general idea may appear straightforward, the magnitude and consistency of the improvements are not obvious without experimentation.** We believe this analysis helps clarify the utility of such approaches and fills an important gap in the early-exit literature.
>
> We will revise the manuscript to better emphasize these points and clarify both our motivation and the practical advantages of the proposed approach.
>
> ## Issue 3
> > I understand that confidence calibration (...) wouldn't it still be important to guarantee calibration?
>
> > Classifiers (...) calibration and EEFP could complementary (...)?
>
> We thank the reviewer for this insightful question. We agree that calibration remains an important property, especially when a given classifier is used to make final predictions in decision-making or cost-sensitive settings. Our work does not contradict the extensive literature demonstrating the value of calibration.
>
> Rather, the point of our work is more specific: **calibration alone is not sufficient to optimally select the classifier to use within an early-exit neural network**. Calibration reflects the correctness of a classifier’s own predictions, but does not indicate whether a deeper classifier would correct its errors. In this sense, calibration and EEFP are complementary. Calibration ensures reliable predictions at a given exit, while EEFP captures the ability to make effective routing decisions across exits.
>
> **Due to the response length limit we refer the reviewer to our response to reviewer FUbk, Issue 1, for details.** We will clarify this point in the revised manuscript.
>
> ## Issue 4
> > If the top-k (...) wouldn't it make sense to model multiclass or class-wise calibration (...)?
>
> As suggested by the reviewer, **we extend our evaluation and compare our method to other calibration approaches.** The results are available at:
>
> https://anonymous.4open.science/r/Rethinking-Calibration-ICML-2026/results_comparison_with_calibration.pdf
>
> Due to the response length limit, **we refer the reviewer to our response to reviewer NEef, issue 2, where we address the same concern, for details.**

---

> > ### Author Rebuttal · Reviewer_qAdB · 2026-04-02
> >
> > My questions have been answered. I already was recommending the paper to be accepted, so I didn't change my score.

---

### Decision · Program_Chairs · 2026-04-30

**Decision:**

Accept (regular)

**Comment:**

This work gives an insightful theoretical argument showing that early exit based on calibration is not ideal and proposes a new method to address this limitation. Reviewers appreciate the novelty and significance of the submission. Additional results included in the rebuttal have led to more positive assessment by the reviewers. I recommend acceptance and trust the authors to include the results from the rebuttal in the camera-ready version.